# DAXX drives de novo lipogenesis and contributes to tumorigenesis

Iqbal Mahmud[1,2,3,4], Guimei Tian[1], Jia Wang[1,5], Tarun E. Hutchinson[1], Brandon J. Kim[1], Nikee Awasthee[1], Seth Hale[1], Chengcheng Meng[1], Allison Moore[1], Liming Zhao [1], Jessica E. Lewis[1], Aaron Waddell[1], Shangtao Wu[1], Julia M. Steger [1], McKenzie L. Lydon [1], Aaron Chait[1], Lisa Y. Zhao[1,8], Haocheng Ding [6], Jian-Liang Li [7], Hamsa Thayele Purayil[1], Zhiguang Huo [6], Yehia Daaka[1], Timothy J. Garrett[2,3] & Daiqing Liao [1] ✉

Cancer cells exhibit elevated lipid synthesis. In breast and other cancer types, genes involved in lipid production are highly upregulated, but the mechanisms that control their expression remain poorly understood. Using integrated transcriptomic, lipidomic, and molecular studies, here we report that DAXX is a regulator of oncogenic lipogenesis. DAXX depletion attenuates, while its overexpression enhances, lipogenic gene expression, lipogenesis, and tumor growth. Mechanistically, DAXX interacts with SREBP1 and SREBP2 and activates SREBP-mediated transcription. DAXX associates with lipogenic gene promoters through SREBPs. Underscoring the critical roles for the DAXX-SREBP interaction for lipogenesis, SREBP2 knockdown attenuates tumor growth in cells with DAXX overexpression, and DAXX mutants unable to bind SREBP1/2 have weakened activity in promoting lipogenesis and tumor growth. Remarkably, a DAXX mutant deficient of SUMO-binding fails to activate SREBP1/2 and lipogenesis due to impaired SREBP binding and chromatin recruitment and is defective of stimulating tumorigenesis. Hence, DAXX's SUMO-binding activity is critical to oncogenic lipogenesis. Notably, a peptide corresponding to DAXX's C-terminal SUMO-interacting motif (SIM2) is cell-membrane permeable, disrupts the DAXX-SREBP1/2 interactions, and inhibits lipogenesis and tumor growth. These results establish DAXX as a regulator of lipogenesis and a potential therapeutic target for cancer therapy.

Cancer cells exhibit elevated intracellular lipid synthesis, resulting in increased levels of fatty acids (FAs), membrane phospholipids, and cholesterol[1,2]. Notably, normal non-proliferating cells generally rely on the uptake of lipids from the circulation for homeostasis. In contrast, highly proliferative cancer cells show strong avidity to acquire elevated lipids and cholesterol through either enhancing the uptake of exogenous (or dietary) lipids and lipoproteins or activating their endogenous lipid synthesis mechanisms, including de novo lipogenesis

[1]Department of Anatomy and Cell Biology, UF Health Cancer Center, University of Florida College of Medicine, Gainesville, FL, USA. [2]Southeast Center for Integrated Metabolomics, Clinical and Translational Science Institute, University of Florida, Gainesville, FL, USA. [3]Department of Pathology, Immunology and Laboratory Medicine, University of Florida College of Medicine, Gainesville, FL, USA. [4]Department of Bioinformatics and Computational Biology, University of Texas MD Anderson Cancer Center, Houston, TX, USA. [5]The Affiliated Cancer Hospital of Zhengzhou University & Henan Cancer Hospital, 450008 Zhengzhou, Henan, China. [6]Department of Biostatistics, University of Florida, Gainesville, FL, USA. [7]Integrative Bioinformatics, National Institute of Environmental Health Sciences, Research Triangle Park, NC, USA. [8]Present address: Department of Medicine, University of Florida College of Medicine, Gainesville, FL, USA. ✉e-mail: dliao@ufl.edu

from non-lipid sources such as glucose and acetate[1–4]. Increased lipid production in cancer cells is thought to supply lipids for the synthesis of membranes and signaling molecules during rapid cell proliferation and tumor growth, due to limited availability of lipids in the tumor microenvironment[1,5]. Lipid synthesis is controlled by several transcription factors, such as the sterol regulatory element-binding proteins, SREBP1 and SREBP2 (SREBP1/2), that have been shown to play an important role in maintaining lipid synthesis in cancer[6]. SREBP1/2 precursors are sequestered in the endoplasmic reticulum. When sterol supply is low, SREBP1/2 are transported to the Golgi apparatus where they are cleaved by proteases, and their N-terminal domain consisting of a basic helix-loop-helix-leucine zipper DNA-binding domain is then released and imported into the nucleus to promote transcription of genes that contain the sterol regulatory elements (*SREs*) required for lipid synthesis[7–10].

Independently of intracellular lipid levels, oncogenic drivers, including KRAS and PI3K, promote de novo lipogenesis in breast cancer (BC) and other cancer types converging on mTORC1 activation[1,11–13]. mTORC1 promotes S6K1-dependent SREBP1/2 processing[14]. The phosphatidate phosphatase Lipin-1 sequesters mature SREBP1/2 in a subnuclear compartment separated from DNAs, thereby preventing SREBP1/2 from activating gene expression. mTORC1 directly phosphorylates Lipin-1, which inhibits its nuclear translocation and thus restores SREBP activity[15]. mTOR signaling also indirectly stabilizes SREBP1/2 by opposing phosphorylation-dependent ubiquitination of SREBP1/2 by the E3 ubiquitin ligase FBXW7 and subsequent proteasomal degradation[16–18]. mTORC1 is shown to phosphorylate the acetyltransferase and transcription coactivator p300 to enhance its acetyltransferase activity, thereby increasing lipogenic gene expression and lipogenesis[19]. Notably, tumors efficiently convert acetate to acetyl-CoA[20], which is predominantly used for lipid synthesis[21], highlighting the need for cancer cells to produce lipogenic enzymes[22]. While the dependence on de novo lipogenesis in cancer is well documented, the mechanisms that control SREBP-mediated transcription underlying oncogenic de novo lipogenesis remain poorly understood.

DAXX, originally discovered as a context-dependent regulator of cell death or survival[23–25], has an extensively documented role in transcription regulation through interacting with transcription factors, including p53[26] and NF-κB[27]. We and others have reported that DAXX is a small ubiquitin-related modifier (SUMO)-binding protein via two SUMO-interacting motifs (SIMs) and that the SUMO-binding property of DAXX appears critical for it to regulate transcription[25,28–30]. More recent studies have defined DAXX as a specific chaperone for the histone variant H3.3[31–33] that may have important roles in regulating chromatin structure, including chromatin binding of transcription factors such as p53[34]. Interestingly, DAXX is shown to act as an ATP-independent chaperone to prevent abnormal protein aggregation[35]. DAXX binds specifically to the H3.3/H4 dimer and deposits it onto chromatin[36,37]. Emerging evidence suggests that DAXX has an oncogenic role in diverse cancer types[38,39], which appears to be linked to its functions in gene regulation[39,40]. Whereas the levels of DAXX expression directly correlate with its ability to promote tumor growth[25,38–41], the molecular mechanisms underlying DAXX's oncogenic function have begun to emerge. Of critical importance is to understand whether DAXX is a tractable therapeutic target for cancer therapy.

In this work, DAXX is identified as a regulator of oncogenic lipogenesis through its interaction with SREBP1/2, leading to activating lipogenic gene expression programs and the promotion of cancer cell proliferation in vitro and tumor growth in vivo. Notably, a SUMO-binding defective DAXX mutant cannot bind SREBP2, stimulate lipogenesis, and accelerate tumor growth. Moreover, a DAXX SIM-derived peptide (SIM2) blocks the DAXX–SREBP interaction, lipogenic gene expression, de novo lipogenesis and tumor growth. Thus, the DAXX SIM/SUMO interface represents a potentially tractable therapeutic

target with the SIM2 peptide. Our study reveals a previously unrecognized oncogenic pathway in BC and suggests a potential therapeutic approach for cancer therapy.

## Results

### DAXX promotes lipogenic gene expression and de novo lipogenesis in a SIM-dependent manner

Bioinformatic analyses of clinical BC samples revealed that *DAXX* mRNA levels are elevated in all four major BC subtypes and correlate with poor prognosis (Supplementary Fig. S1) and BC metastasis (Supplementary Fig. S2)[25]. Consistently, the levels of cholesterol and other lipids are higher in BC samples compared to normal controls[42] (Supplementary Fig. S1c). To understand a potential oncogenic role of DAXX, we used gain- and loss-of-function approaches. We genetically depleted endogenous *DAXX*, overexpressed wild-type (WT) *DAXX*, or a mutant in which both SIM1 and SIM2 (I7K/I733K or DSM) required for binding to SUMO are disabled[28] in the triple-negative BC (TNBC) cell line MDA-MB-231 (Fig. 1a). Microarray-based RNA profiling and Ingenuity Pathway Analysis (IPA) of differentially expressed genes in cells with *DAXX* mRNA knockdown (KD) and WT *DAXX* overexpression (OE) in comparison to control cells revealed a marked downregulation and upregulation, respectively, of the de novo lipogenesis pathway. Remarkably, *DSM* OE failed to activate the lipogenesis pathway (Fig. 1b–d). Lipogenesis regulators (*SREBF1/2* encoding SREBP1/2 and SCAP) were among top inhibited upstream regulators in the KD cells, while WT *DAXX* OE activated *SREBF1/2* (Fig. 1c). Concordantly, the cholesterol biosynthesis via the mevalonate pathway were among the top pathways identified by IPA (Fig. 1d). Most of the genes in the core cholesterol biosynthesis pathway are affected by *DAXX* expression levels (Fig. 1b). Notably, expression of these genes was either unaffected or moderately reduced in cells with *DSM* OE (Fig. 1b). Gene Set Enrichment Analysis (GSEA) of mRNA microarray data confirmed suppression and activation of the de novo lipogenesis pathway by *DAXX* KD and WT OE, respectively (Supplementary Fig. S3a). Of note, several transcriptional regulators that are known to interact with DAXX such as JUN and PML[25] were also affected by DAXX expression levels. Interestingly, the insulin receptor (INSR) pathway that regulates intracellular lipid production[43] also seems to be positively regulated by DAXX (Fig. 1c).

RT-qPCR data for several genes provided validation for the microarray results (Fig. 1e). The impact of *DAXX* KD or OE on lipogenic gene expression was further validated by immunofluorescence microscopy and immunoblotting (Supplementary Fig. S3b, c). Using a tetracycline-inducible gene expression system, we found that *DAXX* induction increased lipogenic gene expression (Supplementary Fig. S3d), providing further evidence that DAXX directly activates lipogenic gene expression. In keeping with our findings, our analysis of public gene expression datasets based on human and mouse cells[44–46] indicated that DAXX is involved in promoting the SREBP/lipogenesis pathway (Supplementary Fig. S4). Further analyses of clinical BC data indicate that *DAXX* expression levels positively correlate with that of *SREBF1* and *SREBF2* (Supplementary Fig. S4d) and a panel of lipogenic genes (Supplementary Fig. S1e). Our previous analysis indicated that *DAXX* is upregulated in BC metastases[25]. As FA synthesis[47] and more broadly the SREBP1 lipogenesis pathway enhance BC brain metastasis[48], we analyzed mRNA levels of *DAXX* and lipogenic genes in BC metastases in different distant organs and found that *DAXX* and genes of selected lipogenic enzymes are consistently and significantly upregulated in BC brain metastases (Supplementary Fig. S2). Collectively, these data provide evidence that DAXX may have an oncogenic function by promoting lipogenic gene expression and that the integrity of both DAXX SIMs appears critical for this property.

### DAXX interacts with SREBP1 and SREBP2

SREBP1/2 are master transcription factors that promote lipid production when the intracellular levels of lipids/sterols are low[6].

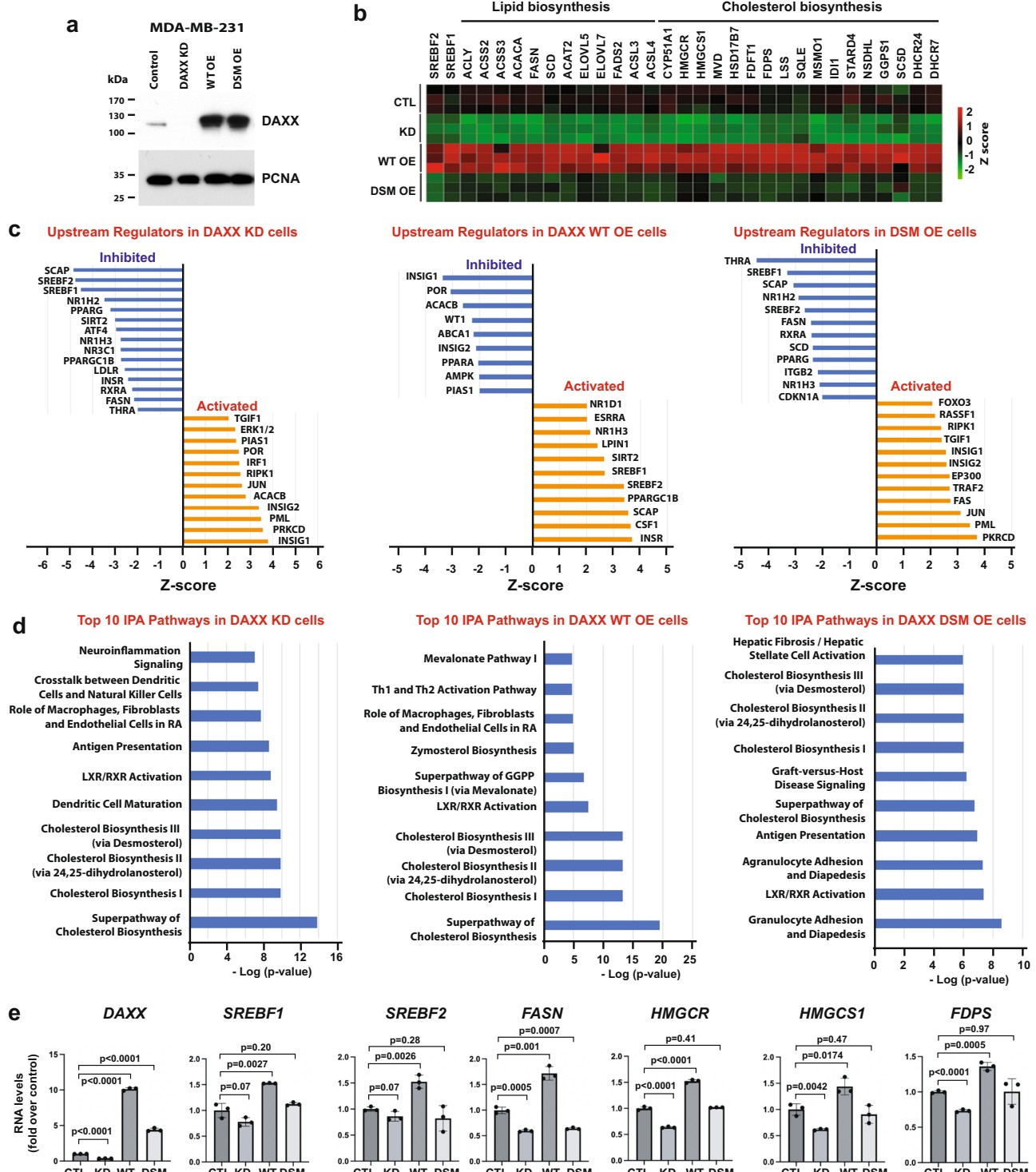

**Fig. 1 | DAXX promotes lipogenic gene expression and its SUMO-binding property is required. a** Validation of shRNA-mediated *DAXX* knockdown (KD) and the overexpression of wild-type *DAXX* (WT OE) and the SUMO-binding defective mutant (*DSM* OE) compared to cells with a control vector (CTL) in MDA-MB-231 cells by immunoblotting. **b** An expression heatmap of selected genes in the lipogenesis pathway in MDA-MB-231-derived cells (CTL, KD, wt OE, and DSM OE) based on microarray data (triplicates for each group). **c, d** Ingenuity pathway analysis (IPA) for upstream regulators (**c**) and canonical pathways (**d**) using differentially expressed genes in KD, WT, and DSM OE cells compared to CTL cells based on the mRNA microarray data as in (**b**). *P* values were obtained via the right-tailed Fisher's exact test implemented in the Ingenuity Pathway Analysis software in (**d**). **e** RT-qPCR analysis of the indicated genes in the MDA-MB-231-derived cells. mRNA level was normalized against that of *ACTB*. Data are shown as mean ± SEM (*n* = 3 biologically independent samples). *P* values are based on unpaired, two-tailed *t* test vs CTL. GGPP Geranylgeranyl diphosphate, RA rheumatoid arthritis. Source data are provided as a Source Data file.

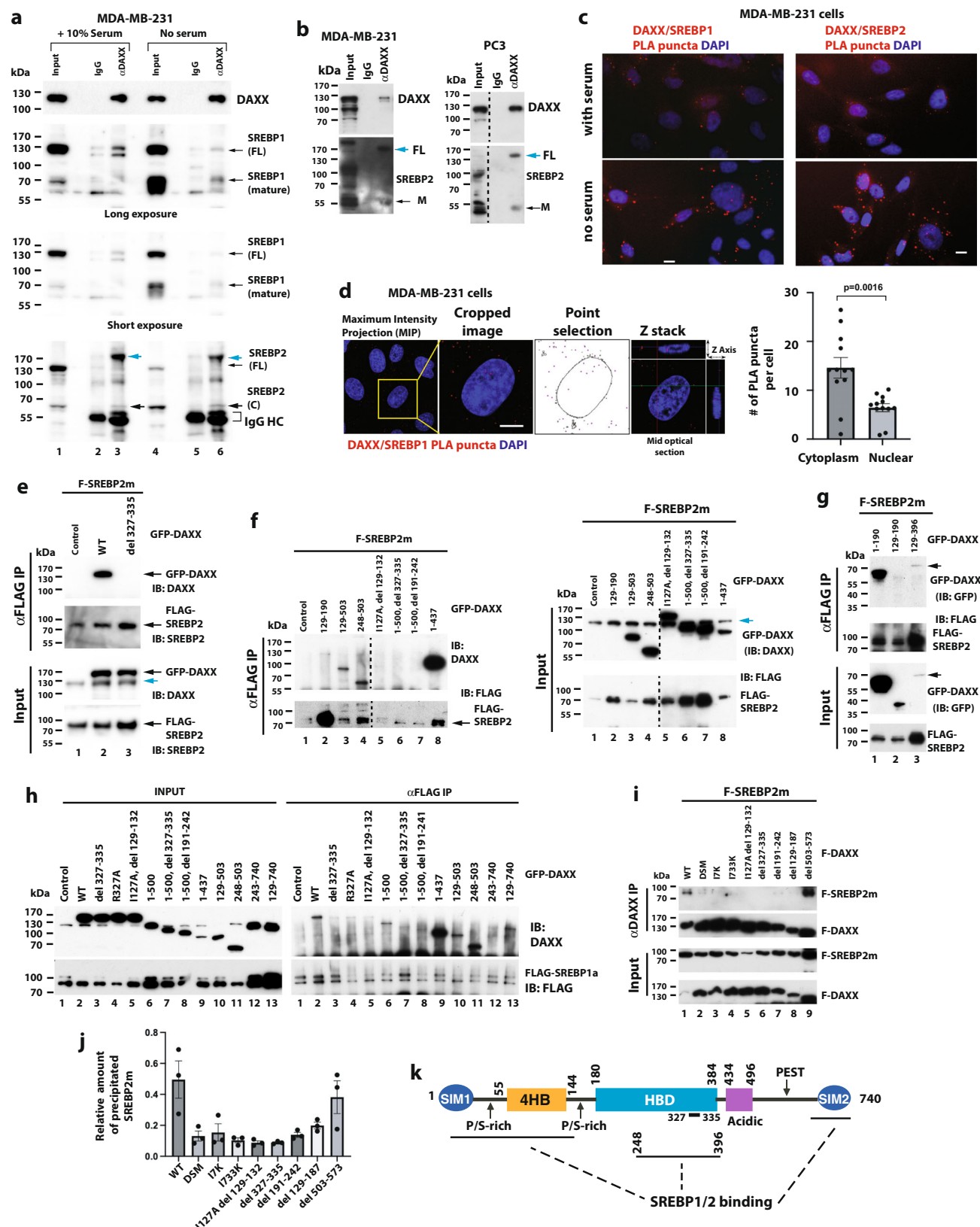

Because DAXX expression levels positively correlate with the activation of the SREBP/lipid biosynthesis pathway (Fig. 1), we reasoned that DAXX could regulate lipid biosynthesis through interacting with SREBP1/2. Immunoprecipitation (IP) of total cell extracts with an anti-DAXX antibody evidenced co-precipitation of two SREBP1/2 proteins that correspond to the SREBP precursor (full-length or FL) and

cleaved mature (M) forms (Fig. 2a, b). Note that the apparent molecular weight (MW) of the SREBP2 precursor co-precipitated with DAXX appears larger than the calculated MW of the reference FL SREBP2 (~125 kDa) and this was detected by two different anti-SREBP2 antibodies (Fig. 2a, b). The expression of a larger SREBP2 isoform has been validated in multiple cancer cell lines

**Fig. 2 | DAXX binds to SREBP1 and SREBP2 and the DAXX SIMs are important for DAXX-SREBP2 interaction. a, b** The endogenous DAXX and SREBP1/2 interact. Total cell extracts of the indicated cell lines were subjected to IP. FL full-length (precursor), M mature, C cleaved C-terminal fragment. **c** Representative images of Proximity Ligation Assay (PLA) showing the interactions of DAXX with SREBP1 and SREBP2. **d** Quantification of nuclear and cytoplasmic DAXX–SREBP1 PLA puncta using confocal microscopy. Data are presented as mean values ± SEM ($n = 12$), and the $P$ value was calculated based on unpaired two-tailed $t$ test. **e–g** The binding sites in DAXX for mature SREBP2. The cell lysates of transfected 293T cells were subjected to anti-FLAG IP. The DAXX constructs were detected with an anti-DAXX antibody in (**e, f**) or an anti-GFP antibody in (**g**). The endogenous DAXX in the input samples is denoted with a cyan arrow in (**e, f**). The arrow points to the GFP-DAXX 129–396 band in (**g**). HC IgG heavy chain. **h** Cotransfection of FLAG-SREBP1a

(mature) and the indicated GFP-DAXX constructs, IP and immunoblotting were performed as in (**e–g**). **i** Both DAXX SIM1 and SIM2 are important for binding to mature SREBP2. The indicated FLAG-tagged DAXX (F-DAXX) and mature SREBP2 (F-SREBP2m) were expressed in 293T cells by transient transfections and subjected to IP with an anti-DAXX antibody. **j** Quantification of SREBP2m co-immunoprecipitated with DAXX. Data are presented as mean values ± SEM ($n = 3$). The band intensities of immunoprecipitated F-SREBP2m and F-DAXX were quantified using the ImageJ software. The band intensities of F-SREBP2m were normalized to the co-precipitated F-DAXX. **k** Schematic drawing of DAXX-SREBP2 interactions. The position of amino acid (aa) 327–335 within the DAXX HBD critical for the DAXX–SREBP interactions is indicated. SIM SUMO-interacting motif, 4HB DAXX helical bundle, HBD histone-binding domain, P/S proline/serine, PEST proline, glutamic acid, serine, and threonine-rich sequence. Numbers refer to aa

(Supplementary Figs. S5 and S6) and in the literature[49,50], possibly due to differential posttranslational modifications. Endogenous as well as exogenous DAXX was co-immunoprecipitated with transiently expressed FLAG-tagged mature SREBP1a, SREBP1c, or SREBP2, as well as transfected FL SREBP1 and SREBP2 (Supplementary Fig. S6a, b). Using Proximity Ligation Assay (PLA) with a mouse monoclonal anti-DAXX and a rabbit polyclonal anti-SREBP1 or SREBP2 antibody, endogenous DAXX–SREBP1/2 interaction signals were detected in both the cytoplasm and the nucleus of the MDA-MB-231 cells (Fig. 2c, d). Interestingly, the number of DAXX/SREBP1/2 PLA signals was significantly increased in the absence of serum (Fig. 2c), suggesting that low extracellular supply of lipids might enhance the DAXX–SREBP1/2 interaction.

Using various *DAXX* deletion constructs in transfected 293T cells, we found that the mature SREBP2 protein interacted with two separate regions of DAXX, the N-terminal part encompassing the well-folded helical bundle termed 4HB (DAXX helical bundle)[51] and a part of the central histone-binding domain (HBD)[36] (Fig. 2e–g). Interestingly, although 4HB and HBD individually bound robustly to SREBP2 (Fig. 2f lanes 3 and 4 and Fig. 2g lanes 1 and 3), the integrity of both binding sites in the context of the full-length DAXX or a longer construct appeared critical for the DAXX-SREBP2 interaction. Indeed, mutations within 4HB (I127A, *del 129–132*) (Fig. 2f, lane 5) or HBD (*del 327–335*, Fig. 2e, lane 3 and Fig. 2f, lane 6; *del 191–242*, Fig. 2f, lane 7) abolished the DAXX-SREBP2 interaction. Notably, the DAXX construct (aa 1–437) lacking the sequence from the acidic domain to the C-terminus seemed to show higher affinity to SERBP2 (Fig. 2f, lane 8). Deletions of C-terminal regions of DAXX did not affect the DAXX-SREBP2 interaction (Fig. 2f, g). The mature SREBP1a bound to DAXX in a similar fashion (Fig. 2h). Immunofluorescence microscopy revealed DAXX/SREBP1 colocalization (Supplementary Fig. S7). Significantly, the I7K or I733K mutations individually or in combination (I7K/I733K or DSM) markedly attenuated the interactions between DAXX and mature SREBP2 (Fig. 2i, lanes 2–4 and Fig. 2j). Collectively, our data demonstrate that the mature SREBP1/2 specifically interact with DAXX via DAXX's 4HB and HBD, and that both SIMs of DAXX are important for the DAXX–SREBP interactions (Fig. 2k).

## DAXX promotes lipid production

To assess the functional impact of DAXX on lipogenesis, de novo lipogenesis assays using [$^{14}$C]-acetate metabolic labeling were conducted. Data shown in Fig. 3a confirmed that *DAXX* expression levels positively correlate with levels of intracellular lipid synthesis, with reduced or increased lipid levels in *DAXX* KD or WT OE MDA-MB-231 cells, respectively. However, the *DSM* mutant did not alter de novo lipogenesis (Fig. 3a). Consistently, mass spectrometry (MS)-based lipidomic profiling revealed that *DAXX* KD reduced, but WT OE increased, levels of specific lipid molecules. The *DSM* mutant was impaired to promote lipid production compared to WT *DAXX* (Fig. 3b and Supplementary Fig. S8a). *DAXX* depletion through CRISPR/Cas9 also reduced lipid production in MDA-MB-231 cells, providing an

independent validation (Supplementary Fig. S8b). To determine a broader role of DAXX in lipogenesis in cancer, we depleted endogenous *DAXX* or overexpressed *DAXX* in several cancer cell lines. In the TNBC MDA-MB-468 cell line, *DAXX* expression levels positively correlated with lipid production based on [$^{14}$C]-acetate labeling experiments as well as lipidomic profiling. Again, the *DSM* mutant displayed a loss-of-function phenotype in lipogenesis (Fig. 3c–e and Supplementary Fig. S8c). Diminished de novo lipogenesis upon *DAXX* depletion was also observed in BC cell lines of luminal subtypes (MCF7 and T47D) and the colon cancer cell line HCT116 (Supplementary Fig. S8d). The impact of DAXX on lipogenesis is summarized in Fig. 3f.

## SREBP-binding sites are enriched in DAXX-associated chromatins

The data presented above suggest that DAXX promotes SREBP-mediated transcription to stimulate lipogenesis. To test this idea, we conducted luciferase reporter assays. As shown in Fig. 4a, forced expression of mature SREBP2, SREBP1a, and SREBP1c by transient transfection increased the activity of the luciferase reporter that is under the control of the *SREBF2* promoter containing a canonical *SRE*. Co-expression of *DAXX* further increased the luciferase activity, while DAXX alone had only minimal effects. In contrast, the *DAXX DSM* mutant was largely defective to coactivate SREBP1/2-mediated transcription (Fig. 4a).

We surveyed genome-wide occupancy of DAXX using chromatin immunoprecipitation-sequencing (ChIP-seq) technology. Overexpression of WT *DAXX*, but not the *DSM* mutant, increased DAXX's chromatin association (Fig. 4b, c). Consistent with other studies[44], DAXX primarily bound to sites in introns and intergenic regions with less frequent association with promoters (Fig. 4d). A de novo motif analysis revealed that SREBP-binding elements were significantly enriched in DAXX-associated sites (Fig. 4e).

Notably, the DAXX DSM mutant exhibited reduced overall chromatin association (Fig. 4b, c) as well as diminished occupancy at individual lipogenic genes compared to WT DAXX (Fig. 4f and Supplementary Fig. S9). De novo motif analysis indicated that the SREBP1 motif was less significantly enriched, while the enrichment of the SREBP2 motif was insignificant in anti-DAXX-precipitated chromatins from MDA-MB-231 *DSM* OE cells (Supplementary Fig. S9a). ChIP-qPCR experiments using an antibody against DAXX or the FLAG epitope demonstrated that WT DAXX but not the DSM mutant (both WT and DSM DAXX carrying an N-terminal FLAG epitope tag) was enriched in the promoters of *FASN*, *ACACA* and *SREBF2* (Fig. 4g). Notably, DAXX was not enriched in the 3' UTR of *FASN* and *ACACA* (Fig. 4g), indicating that DAXX is specifically recruited to the promoter regions of lipogenesis genes. In MDA-MB-231 cells depleted of SREBP2, DAXX's recruitment to the promoters of *FASN*, *ACACA* and *SREBP2* was reduced (Fig. 4h), demonstrating that SREBP2 is critical for DAXX to bind the promoters of lipogenic genes. Together with the data showing that the DSM mutant was impaired for binding to SREBP2 (Fig. 2i), these ChIP and the luciferase reporter assay results show that DAXX is recruited to

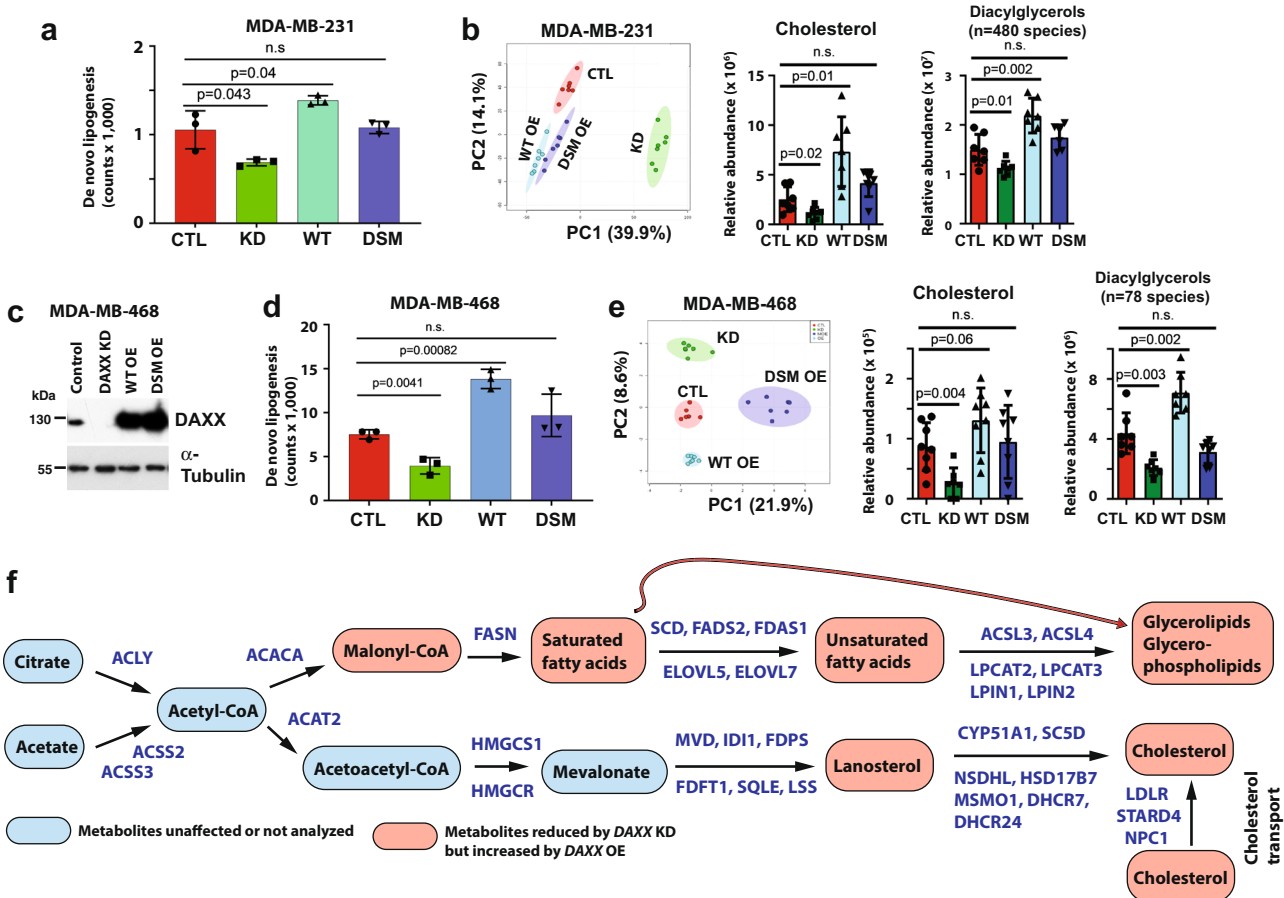

**Fig. 3 | DAXX promotes lipid production. a, d** Impact of *DAXX* expression levels on acetate-dependent de novo lipid synthesis using [$^{14}$C]-acetate labeling in the absence of serum in cell lines derived from MDA-MB-231 (**a**) and MDA-MB-468 (**d**) (*n* = 3 biologically independent samples). **b, e** Principal component analysis (PCA) of lipidome in the panel of four MDA-MB-231 and MDA-MB-468-derived cell lines (CTL, KD, WT OE, and DSM OE). Levels of cholesterol and diacylglycerols as detected by LC/MS in the indicated cell lines are shown. *n* refers to the number of lipid molecules in the indicated lipid classes. **c** Validation of shRNA-mediated *DAXX* knockdown (KD) and the overexpression of wild-type *DAXX* (WT OE) and the SUMO-binding defective mutant (*DSM* OE) compared to cells with a control vector (CTL) in MDA-MB-468 cells by immunoblotting. **f** Lipogenesis pathways activated by DAXX. Representative genes that are downregulated by *DAXX* KD but upregulated by WT *DAXX* OE are shown in blue. Identified metabolites by LC/MS that positively correlate with DAXX expression levels are denoted. All bar graph data are presented as mean values ± SEM. n.s.: not significant (*P* > 0.05, unpaired two-tailed *t* test). Source data are provided as a Source Data file.

the lipogenic gene promoters by SREBP, and that the SUMO-binding activity of DAXX is critical for efficient recruitment of DAXX to the SREBP-bound chromatin. These results suggest a probable molecular explanation for the inability of the DSM mutant to activate lipogenic gene expression and de novo lipogenesis.

### DAXX is critical for tumor growth in vivo

Because DAXX promotes SREBP-mediated gene expression and de novo lipogenesis (Figs. 1–4), and the maintenance of lipid production by SREBPs is critical for cell proliferation and tumor growth[5,52], we hypothesized that DAXX expression levels impact cell proliferation in vitro and tumor growth in vivo. Indeed, *DAXX* KD reduced the number and size of colonies when compared to the control, while WT *DAXX* OE had the opposite effects in the three-dimensional cell culture model of MDA-MB-231. In contrast, the *DAXX* DSM mutant had no effect on either colony size or number (Supplementary Fig. S10).

Next, we examined effects of DAXX expression levels on tumor growth in vivo. In orthotopic BC xenograft models using female mice, *DAXX* KD markedly reduced, while WT *DAXX* OE significantly increased, tumor growth of both MDA-MB-231 and MDA-MB-468 TNBC cell lines (Fig. 5a, b). Notably, the MDA-MB-231 xenograft tumors grew more aggressively than the MDA-MB-468 xenograft tumors (Fig. 5a, b). Notwithstanding, the effects of *DAXX* expression

levels on tumor growth were observed in both TNBC tumor models. The *DSM* mutant OE did not affect tumor growth in either MDA-MB-231 or MDA-MB-468 xenograft models (Fig. 5a, b). Immunoblotting analysis of tumor extracts showed that the levels of the WT DAXX and DSM mutant protein were similar (Fig. 5c), indicating that the inability of the DSM mutant to promote tumor growth was not due to differences in protein expression levels. We profiled the lipids in xenograft tumors derived from cells with different levels of DAXX expression and found that the expression levels of WT DAXX positively correlated with levels of lipids, but the DSM OE had no such effects (Fig. 5d).

In the HCT116 colon cancer xenograft model (both female and male mice) and a prostate cancer xenograft model (male mice), similar tumor growth phenotypes were observed: *DAXX* depletion slowed, while WT *DAXX* OE accelerated, tumor growth (Supplementary Fig. S11a, b). Consistent with the BC xenograft tumor models, *DSM* mutant OE was unable to promote in vivo tumor growth in the colon and prostate cancer xenograft models (Supplementary Fig. S11a, b). In mouse tumor models, *Daxx* depletion also markedly impaired tumor growth in immunocompetent (C57BL/6) and immunodeficient (NSG) mouse hosts (Supplementary Fig. S11c, d). Altogether, our data demonstrate that DAXX exerts an oncogenic property, and that the SUMO-binding activity is critical for DAXX's oncogenic function.

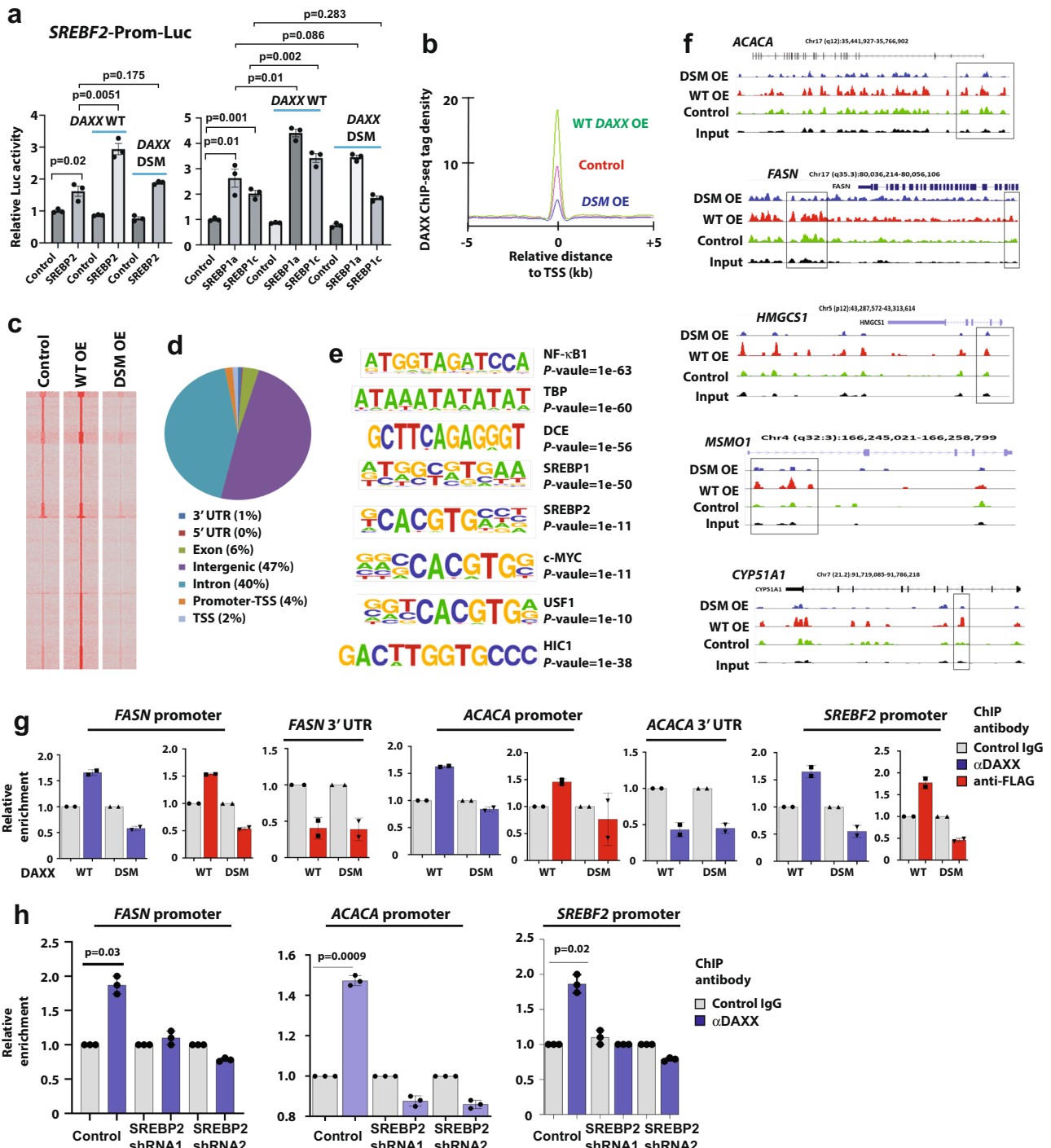

**Fig. 4 | DAXX activates SREBP-mediated transcription and occupies the promoters of lipogenic genes. a** MDA-MB-231 cells were transfected with a luciferase reporter driven by a promoter fragment from the *SREBF2* gene along with mature SREBP2, SREBP1a, SREBP1c, wt DAXX or the DSM mutant cDNA as indicated. Dual luciferase assays were done. Data are presented as mean values ± SEM (*n* = 3 independent transfections). Luc: luciferase. **b**–**f** ChIP-seq analysis of genome-wide occupancy of DAXX. **b**, **c** ChIP-seq signal intensity plot (a comparison of the average DAXX ChIP-seq tag intensities) and heatmaps in MDA-MB-231 control, WT OE, and DSM OE cell lines; signals are centralized to transcriptional start sites (TSS). **d** The genome-wide distribution of DAXX chromatin occupancy. **e** Motifs enriched as determined by the DAXX ChIP-seq dataset of MDA-MB-231 WT OE cells, and (**f**) occupancy of WT and the DSM mutant DAXX in selected lipogenic genes based on ChIP-seq data. Boxes in panel **f** highlight regions near the 5′ end of each gene along with a region 3′ to the *FASN* gene with notable differences in peak heights between

WT DAXX and the DSM mutant. In **e**, the HOMER software uses ZOOPS scoring (zero or one occurrence per sequence) coupled with the hypergeometric enrichment calculations (or binomial) to determine the *P* value for motif enrichment. **g** MDA-MB-231 cells stably expressing WT *DAXX*, and the *DSM* mutant were subjected to ChIP with a control IgG, an anti-DAXX (5G11), or an anti-FLAG antibody (all the DAXX constructs carrying an N-terminal FLAG epitope tag). The precipitated DNAs were subjected to qPCR with primers specific to the promoter regions of the indicated genes. qPCRs for the 3′ UTR of the *FASN* and *ACACA* genes serve as negative controls (*n* = 2 independent ChIP experiments). **h** SREBP2 is critical for DAXX to bind lipogenic gene promoters. MDA-MB-231 cells with a control vector or a SREBP2 shRNA vector were subjected to ChIP with a control IgG, or an anti-DAXX (5G11) antibody followed by qPCR with primers specific to the indicated gene promoters (*n* = 3 independent ChIP experiments). *P* values are based on unpaired two-tailed *t* test. Source data are provided as a Source Data file.

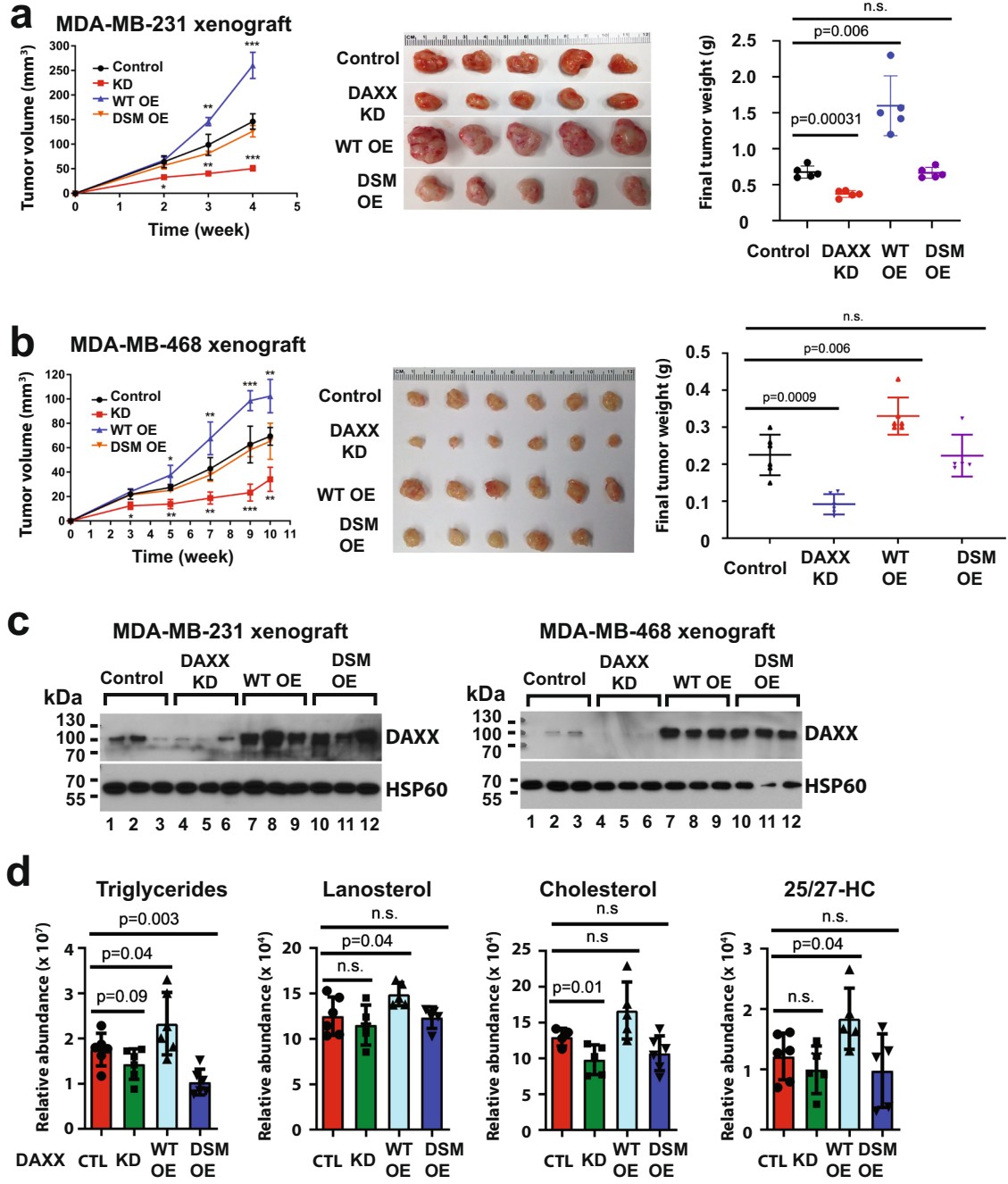

**Fig. 5 | WT DAXX but not the SUMO-binding defective mutant (DSM) promotes tumor growth in vivo. a, b** Cell lines derived from MDA-MB-231 or MDA-MB-468 stably transduced with a control vector (Control), *DAXX* shRNA (KD), wild-type *DAXX* cDNA (WT OE), or the mutant defective of SUMO-binding (*DSM* OE) were implanted into mammary fat pads of female NSG mice. Longitudinal tumor volumes are plotted. Tumor images and weights at the endpoint are shown (*n* = 5 for MDA-MB-231, and *n* = 5 or 6 for MDA-MB-468 xenograft tumors, respectively). **c** DAXX overexpression was maintained for both the WT and DSM mutant proteins.

Protein extracts from three representative xenograft tumors were analyzed for DAXX protein levels using immunoblotting. HSP60 was detected as a loading control. **d** Lipid profiles of xenograft tumors derived from the MDA-MB-231 cell line panel as in (**a**) (*n* = 5 independent tumors). Box plots of representative lipid species in tumor extracts are shown. Data are presented as mean values ± SEM and the *P* values were calculated based on unpaired two-tailed *t* test. \**P* < 0.05; \*\**P* < 0.01; \*\*\**P* < 0.001. 25/27-HC: 25- or 27-hydroxycholesterol. Source data are provided as a Source Data file.

## The DAXX−SREBP axis is important for lipogenesis and tumor growth

SREBP1/2 drive lipid biosynthesis to promote tumorigenesis[6,52]. We have depleted the endogenous *SREBF1/2* or overexpressed the mature form of SREBP1a and SREBP2 in MDA-MB-231 cells (Fig. 6a). *SREBF1/2* KD reduced de novo lipogenesis from acetate (Fig. 6b) and tumor growth in vivo (Fig. 6c, d), whereas the overexpression of mature SREBP1/2 increased lipogenesis and tumor growth (Fig. 6b–d).

Concordantly, lipidomic profiling shows that SREBP1/2 KD had a marked impact on cellular lipidome (Fig. 6e, f). These data indicate that both SREBP1/2 are important mediators of lipogenesis and tumorigenesis. Our data presented above demonstrated that the DAXX−SREBP interactions are critical for lipogenic gene expression, lipid synthesis and tumor growth. To further link SREBP2 to DAXX-mediated tumorigenesis, we depleted *SREBF2* in MDA-MB-231 cells with WT *DAXX* OE. We observed that *SREBF2* KD in the *DAXX* OE cells

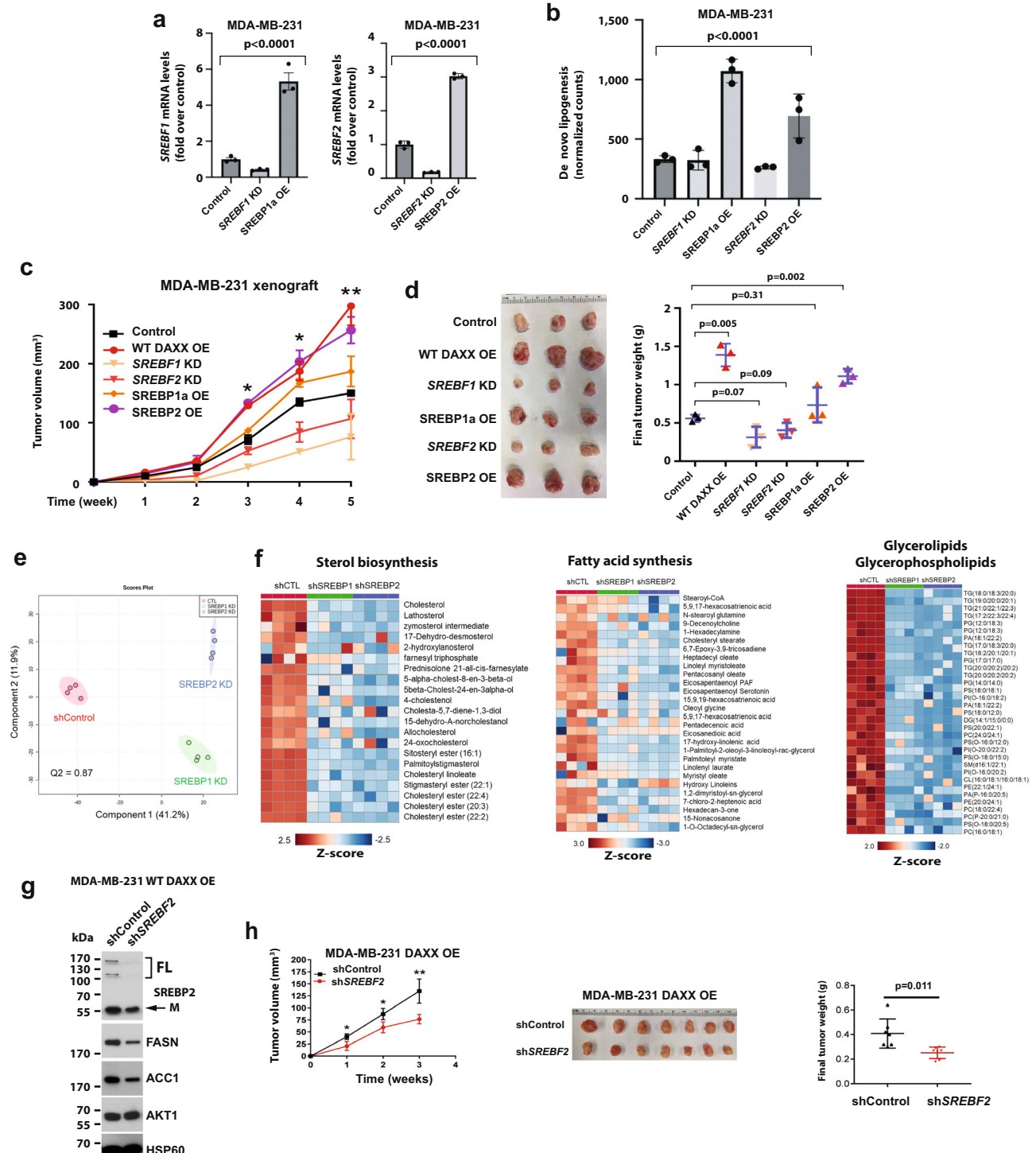

**Fig. 6 | SREBP1, SREBP2, and the DAXX–SREBP axis promote lipid synthesis and tumor growth. a** Cells derived from MDA-MB-231 cell line with a vector for a control, an *SREBF1* shRNA (KD), or SREBP1a (mature) cDNA, an *SREBF2* shRNA (KD), or SREBP2 (mature) cDNA were subjected to RT-qPCR for assessing the expression of *SREBF1* and *SREBF2*. **b** The panel of cell lines in (**a**) were tested for de novo lipogenesis assays using [1-¹⁴C] acetate. In (**a**, **b**), the *P* values were calculated using ordinary one-way ANOVA (*n* = 3 biologically independent samples). **c**, **d** The panel of MDA-MB-231-derived cell line shown in (**a**) were xenografted into mammary fat pads of female NSG mice (*n* = 3 tumor-bearing mice). Tumor volumes were plotted against time (**c**). The images of dissected tumors are shown, and the final tumor weights are plotted (**d**). **e** PCA of lipidomes in shControl (shCTL), shSREBP1 and

shSREBP2 cells. Each dot represents a sample (*n* = 4). **f** Heatmap analysis of selected lipid molecules in the sterol, fatty acid and glycerolipid/glycerophospholipid class of lipids in SREBP1 KD or SREBP2 KD cells compared to control cells. **g** Control or SREBP2 shRNA were expressed in MDA-MB-231 cells with *DAXX* OE. The levels of the indicated proteins were assessed by immunoblot. **h** The indicated cells shown in (**g**) were xenografted into mammary fat pads of female NSG mice. Tumor volumes were plotted against time. Representative images of dissected tumors are shown. The final tumor weights are plotted (*n* = 7 tumor-bearing mice). All data are presented as mean values ± SEM. The *P* values in (**c**, **d**, **h**) were calculated based on unpaired two-tailed *t* test. \**P* < 0.05; \*\**P* < 0.01. Source data are provided as a Source Data file.

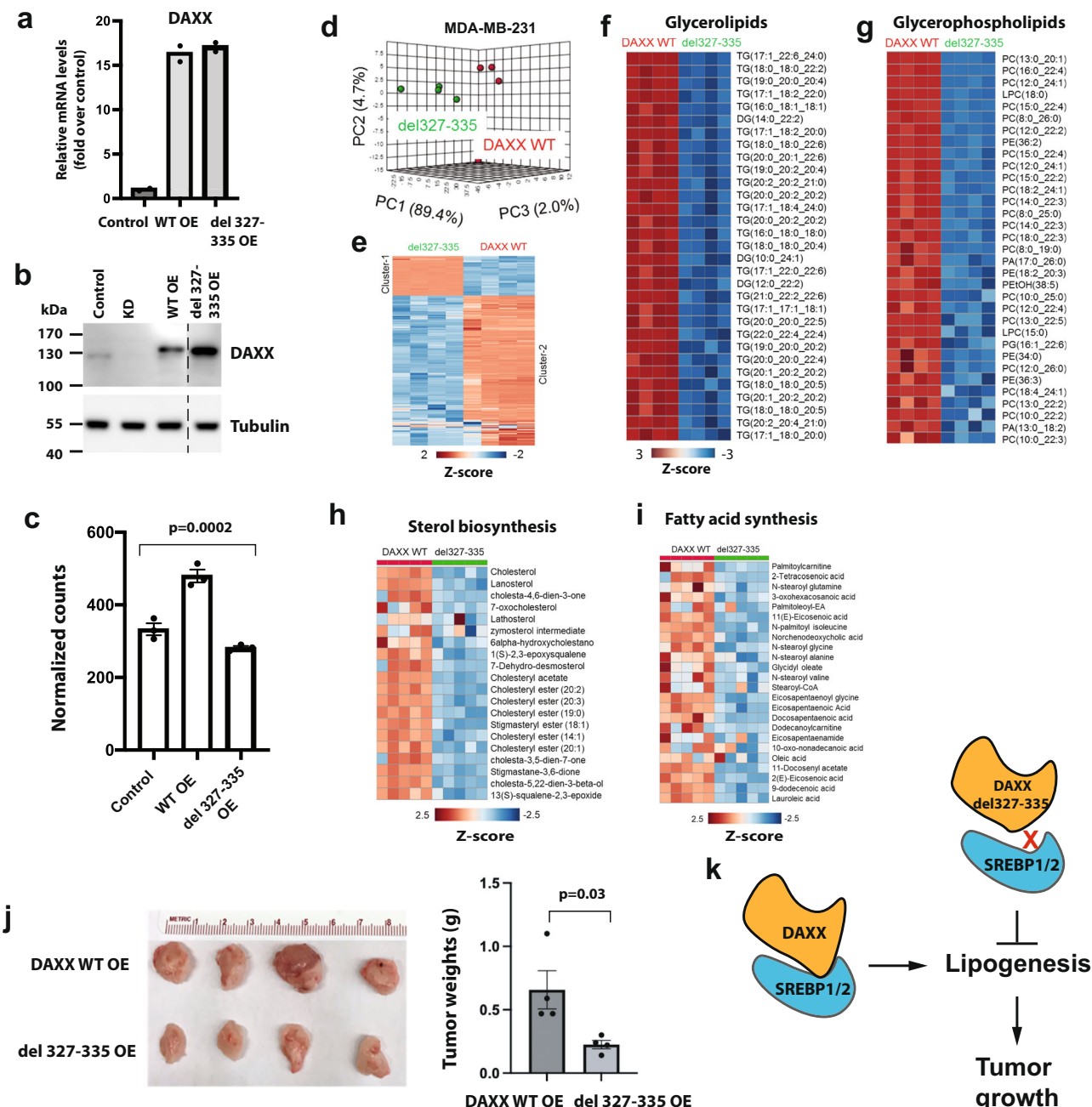

**Fig. 7 | The DAXX–SREBP interaction is critical for lipid synthesis and tumor growth. a** Relative mRNA levels of *DAXX* in MDA-MB-231 cells expressing the WT or *del 327–335* mutant cDNA of *DAXX* as determined by RT-qPCR (*n* = 2). **b** DAXX protein levels in control cells and those with KD, WT and *del 327–335* mutant cDNA of *DAXX*. **c** The *DAXX del 327–337* mutant impaired de novo lipogenesis. Serum-starved cells were labeled with [1-¹⁴C] acetate and total lipids were isolated. Radioactivity was counted and normalized against total protein level. The *P* value was calculated using ordinary one-way ANOVA (*n* = 3 biologically independent samples). **d** PCA of lipidomes in MDA-MB-231 cells expressing the *del 327–335* mutant and WT *DAXX*. Each dot represents a sample (*n* = 4). **e** Hierarchical clustering heatmap analysis of global lipidomes in cells expressing the *del 327–335*

mutant and WT *DAXX*. **f–i** Heatmap analysis of lipid molecules that were highly differentially expressed between MDA-MB-231 cells with the *del 327–335* mutant and WT *DAXX* (**f**, glycerolipids, **g**, glycerophospholipids, **h**, fatty acids, and **i**, sterols). **j** MDA-MB-231 cells expressing the *del 327–335* mutant and WT *DAXX* were xenografted into mammary fat pads of female NSG mice (*n* = 4 tumor-bearing mice). Representative images of dissected tumors are shown. The final tumor weights are plotted. Data are presented as mean values ± SEM and the P values were calculated based on unpaired two-tailed t test. **k** A cartoon depicting the importance of DAXX–SREBP interaction for lipid production and tumorigenesis. Source data are provided as a Source Data file.

significantly attenuated the levels of lipogenic enzymes and tumor growth (Fig. 6g, h), implying that SREBP2 is a critical effector of DAXX's oncogenic function.

To further assess the importance of DAXX–SREBP interaction on lipid synthesis, we overexpressed a DAXX mutant (del 327–335) defective of SREBP1 or SREBP2 binding (Fig. 2e, h) in MDA-MB-231 cells.

The mRNA and protein levels of both WT *DAXX* and the *del 327–335* mutant were similar (Fig. 7a, b). A de novo lipogenesis assay using [¹⁴C]-acetate labeling indicated that the *del 327–331* mutant attenuated lipogeneses (Fig. 7c). Lipidomic profiling revealed that MDA-MB-231 cells expressing the *del 327–335* mutant has a distinct global lipid profile from that of cells expressing the WT *DAXX* (Fig. 7d–i) and that

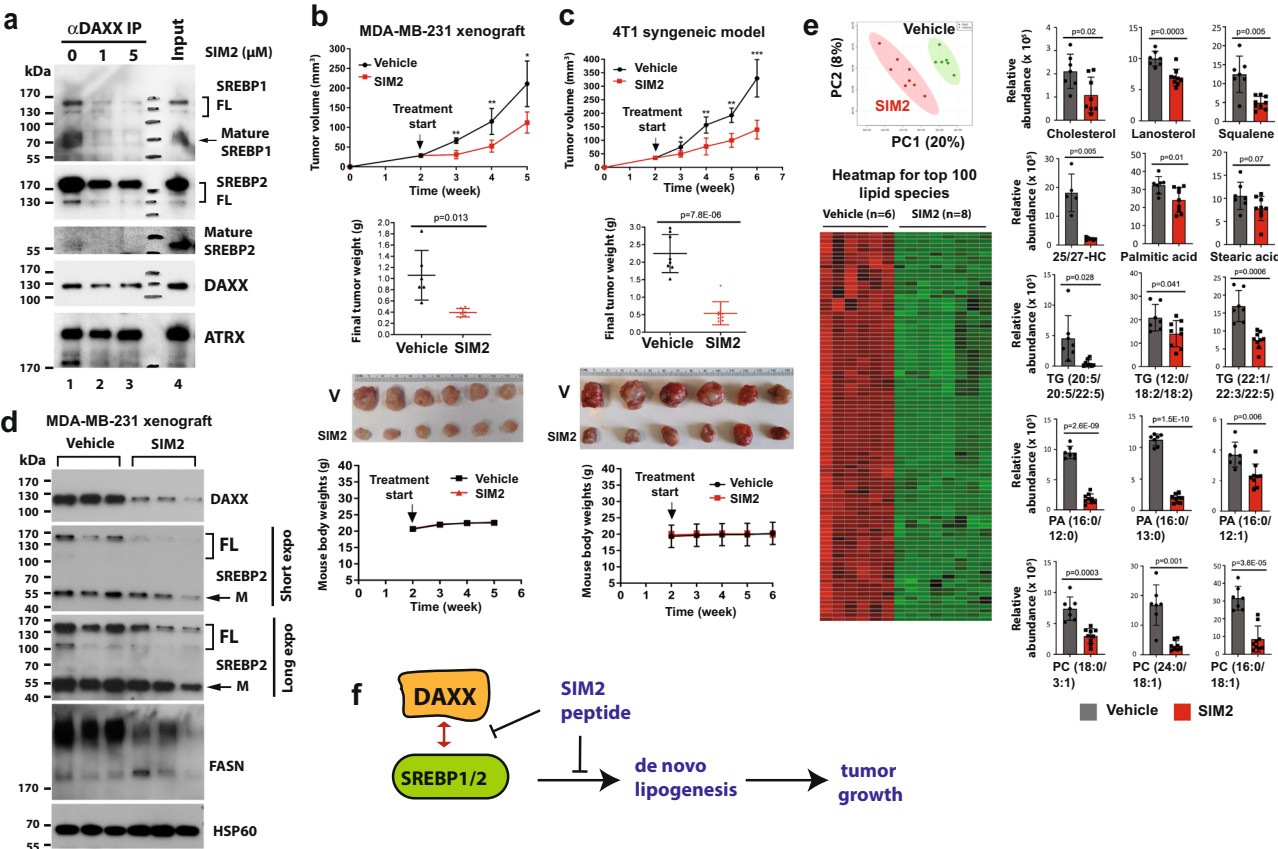

**Fig. 8 | The DAXX SIM2 peptide disrupts the DAXX−SREBP interactions and suppresses tumor growth in vivo. a** MDA-MB-231 cell extracts were subjected to anti-DAXX IP in the absence (0) or the presence of SIM2 (1 and 5 μM). The immunoprecipitates and input were analyzed by immunoblotting. **b** The MDA-MB-231 cells were implanted into mammary fat pads of female NSG mice (*n* = 6 tumor-bearing mice). Mice were dosed i.p. daily weekdays for three weeks with vehicle or SIM2 (25 mg kg⁻¹). Longitudinal tumor volumes, mouse body weights, endpoint tumor weights, and images are shown. **c** The mouse mammary tumor 4T1 cells were injected into mammary fat pads of female BALB/c mice (*n* = 6 tumor-bearing mice). Mice were dosed as in (**b**). Longitudinal tumor volumes, mouse body weights, endpoint tumor weights, and tumor images are shown. **d** SIM2 dosing reduced the expression of DAXX, SREBP2 and FASN in tumors. Protein extracts from three

representative MDA-MB-231 xenograft tumors in vehicle and SIM2-treated mice were subjected to immunoblotting. HSP60 was detected as a loading control. **e** Lipid profiles of MDA-MB-231 xenograft tumors treated with vehicle or SIM2. Shown are a plot of a principal component analysis of lipid molecules detected by LC/MS, a heatmap of top lipid species, and bar graphs of relative abundance of the indicated lipid molecules. Gray and red bars represent tumor samples from mice treated with vehicle or SIM2, respectively. **b, c, e** Data are presented as mean values ± SEM and the *P* values were calculated based on unpaired two-tailed *t* test, \**P* < 0.05; \*\**P* < 0.01; \*\*\**P* < 0.001. V vehicle. **f** A model explaining the antitumor effects of SIM2, which acts to interfere with the interactions between DAXX and SREBP1/2, thereby inhibiting lipogenesis and tumor growth. Source data are provided as a Source Data file.

this mutant was impaired to enhance lipid production, including glycerolipids/glycerophospholipids, sterols, and FAs compared to WT DAXX (Fig. 7f–i). In vivo, the growth of xenograft tumors derived from cells expressing the *del 327–335* mutant was significantly slower than that derived from cells with WT *DAXX* (Fig. 7j). These data collectively indicate that the DAXX−SREBP1/2 interaction is critical for DAXX to promote lipid synthesis and tumorigenesis (Fig. 7k).

## The SIM2 peptide blocks de novo lipogenesis and inhibits tumor growth

DAXX's SUMO-binding property is critical to the DAXX-SREBP2 interaction (Fig. 2), de novo lipogenesis (Fig. 3), DAXX's recruitment to lipogenic genes (Fig. 4), and in vivo tumor growth (Fig. 5). We hypothesized that blocking the interface between DAXX SIMs and SUMOs impairs the DAXX−SREBP interaction, lipid biosynthesis, and tumor growth. To test this hypothesis, we synthesized a peptide corresponding to the C-terminal 12 amino acids of DAXX (amino acids 729–740). This synthetic peptide encompasses the C-terminal SIM of DAXX (termed SIM2). Unexpectedly, SIM2 rapidly internalized into cells (Supplementary Fig. S12a), suggesting that it is a cell-membrane

permeable peptide that does not require specific modifications such as hydrocarbon staple or attaching to a cell-penetrating sequence[53]. SIM2 bound to SUMO1 in vitro (Supplementary Fig. S12b) and blocked de novo lipogenesis in various types of cancer cells, including breast (MDA-MB-231, MDA-MB-468 and Hs578T), prostate (R1-AD1 and R1-D567), colon (HCT116), and mouse tumor cells (4T1 and CT26.CL25) (Supplementary Fig. S12c–h). SIM2 also effectively inhibited de novo lipogenesis in MDA-MB-231 WT *DAXX* OE cells (Supplementary Fig. S12c), and the in vitro growth of MDA-MB-231 cells (Supplementary Fig. S12i). Moreover, SIM2 blocked the expression of lipogenic genes, and the inhibition appeared more potent when DAXX expression level was high (Supplementary Fig. S12j).

Mechanistically, SIM2 inhibited the interaction of DAXX with full-length and mature SREBP1/2, while DAXX's interaction with ATRX was largely unaffected (Fig. 8a), suggesting SIM2 specifically weakens the DAXX−SREBP interaction. To test in vivo efficacy of SIM2 on tumor growth, orthotopic human MDA-MB-231 xenograft tumors were generated in the mammary fat pads of female NSG mice. As shown in Fig. 8b, SIM2 significantly impeded tumor growth, and at the experimental endpoint, the tumor masses were significantly smaller in the

SIM2 treatment group compared to the vehicle control group. In a syngeneic mouse mammary tumor model, mouse 4T1 cell line (derived from a spontaneous mouse mammary tumor with human basal/TNBC characteristics[54]) was transplanted into mammary fat pads of female immunocompetent BALB/c mice. 4T1 tumor-bearing mice were dosed with vehicle or SIM2. Consistent with the MDA-MB-231 xenograft results (Fig. 8b), SIM2 effectively inhibited the 4T1 tumor growth (Fig. 8c). SIM2 peptide appeared safe in vivo, as there was no difference in the body weights of mice in the control and SIM2 dosing group (Fig. 8b, c). To understand the potential mechanism of action of SIM2, MDA-MB-231 tumor extracts were subjected to immunoblotting. Interestingly, the protein levels of DAXX, SREBP2, and FASN were lower in the SIM2 treatment group compared to the control group (Fig. 8d), suggesting that SIM2 acts to downregulate the DAXX-driven lipogenesis pathway. Lipidomic profiling of MDA-MB-231 xenograft tumor samples from each treatment group revealed a marked suppression of lipid production by SIM2 (Fig. 8e). Collectively, these data support a model in which SIM2 exerts its antitumor ability by blocking the DAXX−SREBP interaction, lipogenic gene expression, and lipid production (Fig. 8f).

## Discussion

Lipid availability for proliferating cells determines the activity of intracellular lipid biosynthesis pathway. In a nutrient-poor tumor microenvironment, limited supplies of lipids necessitate the activation of intracellular lipid production in tumor cells for sustained tumor growth. An elaborate sterol sensing machinery controls the cleavage and nuclear translocation of SREBP1/2, which promote expression of enzymes required for de novo lipogenesis[1,55]. SREBP1/2 in conjunction with other transcription factors, such as the E-box-binding basic helix-loop-helix (bHLH) transcription factor USF1, activate expression of lipogenic enzymes and regulators[43]. Other coregulators of gene expression such as acetyltransferases (e.g., p300 and PCAF) as well as oncogenic signaling pathways (e.g., KRAS and mTOR) also play important roles in stimulating de novo lipogenesis[12,56]. We demonstrated here that DAXX is a regulator for de novo lipogenesis. Mechanistically, DAXX interacts with SREBP1/2 and is enriched in chromatins containing SRE motifs. Importantly, DAXX mutants that cannot bind SREBP1/2 are unable to promote lipogenesis and tumor growth. SREBP2 downregulation reduced tumor growth driven by DAXX OE. Hence, our data support a model by which DAXX enhances lipogenesis through interacting with SREBP1/2 to promote lipogenic gene expression, lipid production, and tumorigenesis.

Significantly, the SUMO-binding property of DAXX appears important for DAXX's interaction with SREBP2, chromatin recruitment, de novo lipogenesis and tumor growth. In support of this, we found that the SIM2 peptide, corresponding to the C-terminal SIM of DAXX, inhibits de novo lipogenesis and suppresses in vivo tumor growth (Fig. 8 and Supplementary Fig. S12). SIM2 blocks the DAXX−SREBP1/2 interactions, and the treatment with SIM2 reduces lipid levels and downregulation of DAXX, SREBP2, and FASN in vivo (Fig. 8 and Supplementary Fig. S12). Hence, the inhibition of DAXX/SREBP pathway by SIM2 blocks de novo lipogenesis, which may underlie SIM2-mediated antitumor effects. SIM2 has a unique sequence and displays a high affinity to SUMO1 upon phosphorylation[25,29]. Notably, peptides generally do not penetrate cell membranes. However, SIM2 is spontaneously and rapidly internalized into cells (Supplementary Fig. S12a) and is well tolerated by animals (Fig. 8). Overall, our data indicate that the interface between DAXX SIM and SUMO represents a potentially tractable therapeutic target to inhibit de novo lipogenesis for cancer therapy. Our data also highlight the therapeutic potential of SIM2 as a novel class of cancer therapeutics, which has implications for developing peptide-based drugs.

SUMOylation of nuclear proteins such as transcription factors and core histone proteins is extensively documented[57,58], and SUMOs are highly enriched in the core promoters/enhancers of actively transcribed genes[59–61]. Of relevance to this study, SUMOs have been detected in some gene promoters regulated by SREBP1/2 (e.g., ACLY)[60]. Other studies demonstrated that DAXX's SUMO-binding property is essential for DAXX's recruitment to chromatin and histone H3.3 deposition[62,63]. Our data show that SIM2 preferentially disrupted the DAXX−SREBP1/2 interactions with little effects on the DAXX-ATRX interaction (Fig. 8a), and that the DSM mutant could neither bind to mature SREBP2 (Fig. 2i) nor be effectively recruited to chromatins (Fig. 4). Disruption of either of the two SIMs in DAXX weakened its interaction with mature SREBP2 (Fig. 2i). Both SREBP1/2 are SUMOylated at multiple sites[64–66]. Likewise, DAXX is also SUMOylated at a number of lysine residues[25,30,57,58]. However, SREBPs lack recognizable SUMO-binding domains including SIMs. Collectively, our data, suggest that, via its two SIMs, DAXX may bind preferentially to SUMO-modified SREBPs, and that blocking the DAXX SIM/SUMO interface may weaken the DAXX−SREBP interactions, thereby impairing DAXX's recruitment to SREBP-bound chromatin sites. Further studies will be needed to determine the roles of SREBP SUMOylation in DAXX-mediated activation of lipogenic transcription.

Our data demonstrate that DAXX acts to promote SREBP-mediated transcription. It has been well documented that DAXX can activate and repress transcription, depending on coregulators that are associated with DAXX[25]. Epigenetic modifiers such as HDACs and DNA and histone methyltransferases are involved in DAXX-mediated transcription repression, while coactivators (e.g., CBP) are involved in DAXX-mediated gene activation. The H3.3 histone chaperone function of DAXX is also implicated in both transcriptional activation[67,68] and repression[69,70]. Independently of H3.3 deposition by DAXX to chromatin, the H3.3/H4 dimer metabolically stabilizes DAXX protein, which indirectly enhances repression of endogenous retroviruses (ERVs) by a complex consisting of the DAXX-H3.3/H4 sub-complex, HDAC1, KAP1, and SETDB1[46]. Our data show that SREBP1/2 bind to DAXX by contacting both 4HB and HBD (Fig. 2). A previous structural study demonstrates that a peptide within the transactivation domain of p53 binds to DAXX 4HB[51]. It will be interesting to assess whether DAXX also engages the transactivation domain of SREBPs to promote transcription and whether the H3.3 chaperone function of DAXX is important for lipogenic gene expression.

Of note, the binding motifs of other known DAXX-binding transcription factors such as NF-κB[27] were significantly enriched in DAXX ChIP-seq peaks (Fig. 4e). Our data also implicate the chromatin recruitment of DAXX by other transcription factors such as RUNX1, RUNX2, HIC1 and c-MYC that were not previously shown to interact with DAXX (Fig. 4e and Supplementary Fig. S9). Furthermore, DAXX might interact with the core-transcriptional machinery, as the TATA-box and DCE (downstream core element) were enriched in DAXX-binding chromatins (Fig. 4e and Supplementary Fig. S9). These observations suggest a broader role for DAXX in transcription regulation.

DAXX appears to interact with SREBP1/2 in both the cytoplasm and the nucleus (Fig. 2). In the nucleus, the DAXX−SREBP interactions are expected to mediate DAXX's chromatin recruitment and the activation of lipogenic gene expression. The functional effects of DAXX−SREBP interactions in the cytoplasm are currently unknown. In the cytoplasm, DAXX has been shown to interact with regulators of cell death and cell survival[25]. A recent study demonstrates that DAXX promotes the formation of SQSTM1/p62 membrane-less liquid compartments to activate cellular anti-oxidative stress response[71]. Interestingly, the number of DAXX−SREBP1/2 PLA complexes increases upon serum starvation (Fig. 2c), suggesting that a low level of lipid supply may trigger the formation of the DAXX−SREBP1/2 complexes in the cytoplasm. Of note, our data show that the full-length SREBP1/2 were detectable in the nuclear fraction (Supplementary Fig. S5a, b). In earlier studies, the full-length SREBP1/2 were found in purified nuclear

extracts and nuclear membranes[9] and appear diffusely throughout the cell including the nucleus[15]. In agreement with these early observations, our data show that SREBP1 are found diffusely in the cytoplasm and the nucleus (Supplementary Fig. S7). Whereas these data suggest that SREBPs might not be restricted in the ER and Golgi membranes in the cytoplasm, in some cells, higher levels of SREBP1 were seen in perinuclear areas, reminiscent of ER membrane localization (denoted with yellow arrows in Supplementary Fig. S7a, top). Therefore, the DAXX−SREBP interaction as detected by PLA might not just occur on ER (Fig. 2c). In the absence of serum, SREBP1 is markedly enriched in the nucleus (Supplementary Fig. S7b, c). Nonetheless, the determination of precise localization of SREBPs in cytoplasmic structures in addition to ER and Golgi will require biochemical fractionation of various compartments and high-resolution imaging methods such as electron microscopy. Notably, mature SREBPs were found predominantly in the nucleus (Supplementary Fig. S5a, b). Therefore, the cytoplasmic SREBPs are largely SREBP precursors. Our data show that DAXX interacts with both precursor and mature SREBPs. Interestingly, DAXX seems to promote SREBP1 nuclear localization (Supplementary Fig. S7a), and DAXX OE increased the protein levels of mature SREBP1 (Supplementary Fig. S7d). These observations raise an intriguing possibility that the DAXX−SREBP interaction might facilitate SREBP maturation and nuclear translocation of mature SREBPs. Interestingly, mTORC1 inhibition appears to result in a Lipin-1-mediated sequestration of mature SREBPs in nuclear periphery and a subnuclear compartment inaccessible to DNA in NIH3T3 cells[15]. It will be interesting to assess whether DAXX could prevent such sequestration to facilitate SREBPs to bind chromatin. The functional ramification of the interaction between DAXX and full-length precursor SREBPs requires further investigation.

Epidemiological studies suggest beneficial effects of statin use for lowering BC risk for ER-positive (ER +) and ER-negative BC subtypes[72–77]. In vitro, ER-negative and basal-like BC cell lines appear more sensitive to statins than ER + BC cell lines[78,79]. However, whether statin use would benefit patients with specific BC subtypes remains to be established. Significantly, acquired resistance to endocrine therapies in ER + BCs is associated with increased cholesterol production[80,81]. In particular, elevated expression of enzymes involved in cholesterol synthesis (e.g., SQLE) is associated with poor response to aromatase inhibitors in clinical samples[80]. Statins were shown to reduce ERα chromatin binding and invasive growth of ER + BC cells[81]. Our data show that DAXX promotes lipid production in different cancer types in vitro and in vivo. Concordantly, SIM2 was effective to inhibit de novo lipogenesis in ER + BC and TNBC cell lines and effectively inhibited in vivo tumor growth of basal/TNBC tumor models (Fig. 8 and Supplementary Fig. S12). It will be interesting to test whether DAXX is involved in promoting cholesterol synthesis associated with treatment resistance, and whether DAXX inhibition using agents such as SIM2 could sensitize treatment response to statins.

## Methods
Our research complies with all relevant regulations involving recombinant DNAs, chemical hygiene, and management of chemical, biohazardous, and radioactive materials at the University of Florida. All animal protocols for this study were approved by the University of Florida Institutional Animal Care and Use Committee (IACUC).

### Cell culture
Cells were cultured in Dulbecco's Modified Eagle's Medium (DMEM with 4.5 g/L glucose, L-glutamine and sodium pyruvate, Corning) with 10% bovine calf serum (HyClone, GE Healthcare Bio-Sciences, Pittsburgh, PA), penicillin (10 units/ml), and streptomycin (10 μg/ml) (the complete DMEM medium). The T47D cell line was cultured in DMEM plus 10% fetal bovine serum (Atlanta Biologics, Atlanta, GA), penicillin

(10 units/ml), and streptomycin (10 μg/ml). To culture cells in serum starvation condition, the serum-containing medium was removed from cell cultures after overnight incubation and the culture was washed once with phosphate-buffered saline (PBS, without calcium and magnesium, Corning). Cells were then cultured in serum-free DMEM. For culturing cells in suspension (3D culture), plates were coated with a 1:1 mixture of Matrigel (Corning, Tewksbury, MA) and complete DMEM medium. A desirable number of cells were suspended in the Matrigel and medium mixture and layered on the top of the solidified Matrigel. Complete DMEM medium was added after the Matrigel was solidified. Medium was replaced with fresh complete medium every three days. Colonies were imaged under a microscope; colony numbers and sizes were quantified. Human cell lines (MDA-MB-231, MDA-MB-468, Hs578t, MCF7, T47D, HCT116, PC-3, and 293T) were obtained from ATCC (Manassas, VA). R1-AD1 is a subline of the PCa cell line CWR-R1, and R1-D567 is an engineered derivative line of R1-AD1 that expresses C-terminally truncated AR (AR-Vs)[82]. R1-AD1 and R1-D567 were provided by Dr. Scott Dehm, University of Minnesota. The mouse cancer cell lines 4T1, CT26.CL25, and TRAMP-C2 were from ATCC. The mouse BC cell line E0771 was from CH3 BioSystems (Amherst, NY). Human cell lines (MDA-MB-231, MDA-MB-468, Hs578t, MCF7, T47D, PC-3, HCT116) and mouse 4T1 cell line were authenticated by STR profiling at Genetica DNA Laboratories (Burlington, NC). The other cell lines (293T, CT26.CL25, TRAMP-C2, E0771, R1-AD1, and R1-D567) were recently acquired from vendors or academic labs and were not subjected to further authentication.

### DNA constructs
cDNAs for wild-type (WT) DAXX and mutants with a 5' coding sequence for the FLAG epitope tag and a 3' coding sequences for the MYC and 6x His tags were cloned into a lentiviral vector under the control of the cytomegalovirus immediate early (CMV *IE*) promoter. GFP-DAXX constructs were cloned in the pEGFP-C2 vector. A short hairpin RNA (shRNA) targeting the DAXX coding sequence (nucleotide 624–642, 5'-GGAGTTGGATCTCTCAGAA-3') was cloned into a lentiviral vector under the control of the human *H1* promoter. An shRNA construct with a scrambled sequence (Plasmid # 36311) was from Addgene. Expression vectors for mature SREBP1a (Plasmid # 26801), mature SREBP1c (Plasmid # 26802), mature SREBP2 (Plasmid # 26807), full-length SREBP1 (Plasmid # 32017) and SREBP2 (Plasmid # 32018) were purchased from Addgene. The shRNA clones for *SREBF1* (TRCN0000020607 and TRCN0000020605), and *SREBF2* (TRCN0000020667 and TRCN0000020668) were from the human pLKO.1 TRC Library collection at the University of Florida Health Cancer Center. The *SREBF2* shRNA vector TRCN0000020667 was used to knockdown *SREBF2* expression in MDA-MB-231 cells with DAXX OE. A *SREBF2* promoter fragment was PCR amplified from the genomic DNA isolated from MDA-MB-231 cell line and cloned at sites upstream of the firefly luciferase reporter by the Gibson assembly method. The DNA sequence was confirmed by Sanger sequencing. The PCR primers are shown in Table 1.

Stable expression of cDNA and shRNA was established through lentiviral transduction of cell lines and puromycin (2 μg/ml) selection. The derived cell lines were cultured with DMEM without puromycin.

### Microarray, RNA-seq, and qRT-PCR
For microarray experiments, cells were cultured in the complete DMEM or serum-free DMEM, and total RNAs were isolated using the RNeasy kit (Qiagen). The RNAs were then processed for microarray hybridization to the Affymetrix GeneChip Human Transcriptome Array 2.0[83,84]. Probe set files (.cel file) were normalized by RMA algorithm and analyzed using both R statistical package (https://www.r-project.org/) as well as Affymetrix expression and transcriptome console software from ThermoFisher Scientific. Microarray transcriptomic data collected at different times were batch-collected using ComBat 3.0

**Table 1 | PCR primers used for this study**

| RT-qPCR primers (5' to 3') | |
|---|---|
| ActinB-F-Real | GCTCCTCCTGAGCGCAAGTACTC |
| ActinB-R-Real | GTGGACAGCGAGGCCAGGAT |
| Daxx-RT-F | GAGGCGTCTCTCCTCACAAC |
| Daxx-RT-R | TCTCATGCACTGACCTTTGC |
| SREBP1-F | CTGCTGTCCACAAAAGCAAA |
| SREBP1-R | GGTCAGTGTGTCCTCCACCT |
| SREBP2-F | ATCGCTCCTCCATCAATGAC |
| SREBP2-R | TTCCTCAGAACGCCAGACTT |
| FASN-F | CACAGGGACAACCTGGAGTT |
| FASN-R | ACTCCACAGGTGGGAACAAG |
| HMGCR-F | GTCATTCCAGCCAAGGTTGT |
| HMGCR-R | CATGGCAGAGCCCACTAAAT |
| STARD4-F | GGCGAGTTGCTAAGAAAACG |
| STARD4-R | CCCTGGGCGTATATGGTCTA |
| HMGCS1-F | GGGACACATATGCAACATGC |
| HMGCS1-R | CACTGGGCATGGATCTTTTT |
| FDPS-RT-F | CCAAGAAAAGCAGGATTTCG |
| FDPS-RT-R | CCGGTTATACTTGCCTCCAA |
| **ChIP primers** | |
| FASN-prom-F1 | TAGAGGGAGCCAGAGAGACG |
| FASN-prom-R1 | GCTGCTCGTACCTGGTGAG |
| ACACA-prom-F1 | CAAGGGAAATTGAGGCTGAG |
| ACACA-prom-R1 | CGTTCCAGGAGCATCTGATT |
| FASN-ChIP-3'UTR-F1 | GTGAACCATGACTGCACCAC |
| FASN-ChIP-3'UTR-R1 | GAGCCCTCGGTGACATACAT |
| ACACA-ChIP-3'UTR-F1 | TCATGGCCAAACTGTTGAAA |
| ACACA-ChIP-3'UTR-R1 | TGGGGTCCATTGTTTCTGAT |
| SREBF-prom-F2 | TCCTTTAAACAAGGCGGAGA |
| SREBF-prom-R2 | TCAGCAGCTCAGATTTGCAT |
| **Primers for amplifying a fragment in the SREBF2 promoter** | |
| SREBF2-prom-F2 | CAGCTGAAGCTTGCATGCCTGCAGGTAGGCAGCTGGGAAGATGA |
| SREBF2-prom-R2 | GAGTATATATAGGACTGGGGATCCGTGAGGGTCTCCATGGTCTC |

(combining batches) tools from Broad Institute[85]. Different batches with the same numbers of class labels and sets of transcripts were used for batch correction. For RNA-seq, total RNAs were used for poly A library construction and sequencing was done with 50M raw reads/sample using the Illumina NovaSeq 6000 S4 2x150 platform at the Interdisciplinary Center for Biotechnology Research, University of Florida.

For quantitative real-time PCR (RT-qPCR), the isolated RNAs were reverse transcribed with random hexamers using 2 μg of total RNA, an RNase inhibitor, and reagents in the Multiscribe reverse transcriptase kit (Life Technologies). The resulting cDNAs were diluted and used as input for qPCR using the SYBR green detection method using the Applied Biosystems 7500 Fast Real-Time PCR system. The relative levels of gene expression were determined using the ΔΔCt method with the Ct values of *ACTB* expression as the common normalizer. The primers for qPCR and other applications are provided in Table 1.

**Immunoprecipitation (IP) and immunoblotting**
Cell pellets were resuspended in the IP lysis buffer (50 mM Tris-HCl, pH 7.5, 0.5% Igepal-CA630, 5% glycerol, 150 mM NaCl, 1.5 mM MgCl$_2$, and 25 mM NaF) containing 100-fold diluted protease inhibitor cocktail (Millipore-Sigma P8340). The cell suspension was subjected to two freezing/thawing cycles. The cell lysates were then centrifuged at 21,300 × $g$ at 4 °C for 20 min. The supernatant was used for IP with a

control or an antibody to a specific protein at 2 μg per IP in the presence of protein A-agarose beads. The hybridoma supernatant of the anti-DAXX 5G11 mAb clone was used for immunoprecipitating endogenous DAXX. FLAG-tagged constructs were immunoprecipitated with the M2 mAb (Millipore-Sigma F1804). For IPs in the presence of the SIM2 peptide, SIM2 in PBS was added to a desired concentration. IP mixtures were rotated at 4 °C overnight. The beads were washed four times with the IP lysis buffer and once with the RIPA buffer (50 mM Tris-HCl, pH 7.4, 150 mM NaCl, 1% NP-40, 0.5% sodium deoxycholate). The beads were resuspended in the IP lysis buffer along with one fifth of the volume of the 6x SDS sample buffer (0.375 M Tris-HCl, pH 6.8, 12% SDS, 60% glycerol, 0.6 M DTT, and 0.06% bromophenol blue). Samples were heated at 95 °C for 5 min and chilled on ice for 2 min. After brief centrifugation, the samples were loaded on a 4–20% gradient gel (Novex Tris-Glycine Mini Gels, ThermoFisher). Proteins were then electrotransferred to an Immobilon®-P polyvinylidene fluoride (PVDF) membrane (Millipore). The membrane was blocked with 5% nonfat milk, incubated with a primary antibody and a proper secondary antibody. In Fig. 2a, the rabbit polyclonal anti-SREBP1 antibody (Proteintech 14088-1-AP) and the mouse mAb (BD Bioscience, 557037) were used for detecting SREBP1 and SREBP2, respectively, while the rabbit polyclonal anti-SREBP2 antibody (Cayman Chemical Company, 10007663) was used to detect SREBP2 in Fig. 2b. The proteins were detected using a chemiluminescent

detection kit (Millipore) by exposure to the Fuji Super RX-N X-ray films or an Imager (GE Amersham 680).

For the IP experiments shown in Fig. 2i, the proteasome inhibitor MG132 (10 μM) was added to the cell culture of transfected 293T cells at 3 h before cell lysis. The whole cell extracts of the transfected cells were subjected to IP with a rabbit polyclonal antibody against DAXX (custom-made in this study). The DAXX constructs and F-SREBP2m were detected with the mouse monoclonal anti-FLAG antibody M2 (Millipore-Sigma F1804). Western blot band intensity was quantified with ImageJ 1.53a and analyzed with Microsoft Excel for Mac.

For immunoblotting analyses of cell lysates of monolayer cultures, medium was removed from culture plates and RIPA buffer was added. The plates were frozen at −80 °C overnight and then thawed at room temperature. The lysates were transferred to a centrifuge tube. To prepare tumor lysates, xenograft tumor tissues were fragmented in the presence of liquid nitrogen, ~50 mg of tumor fragment was homogenized in 1 mL of 1× RIPA lysis buffer on ice using a micro-homogenizer. After brief sonication at a low power output for 5 sec on ice, the lysates were cleared by centrifugation at 21,300 × g for 15 min at 4 °C. Protein contents were quantified using a Qubit protein assay kit. Protein extracts from cell culture or tumor lysates were subjected to SDS-PAGE and electro-blotting as above. The antibodies used for this study are listed below (Table 2).

### Proximity ligation assay (PLA) and immunofluorescence microscopy (IF)

The PLA reagents were obtained from Millipore-Sigma (DUO92101-1KT). The assays were performed following the manufacturer's protocol. The antibodies against SREBP2 (Cayman Chemical Company, 10007663), SREBP1 (ProteinTech, 14088-1-AP), DAXX (5G11 hybridoma supernatant) were used for the PLA and IF experiments. IF experiments were performed as described previously, and images were acquired with a Zeiss AxioPhot microscope equipped with an Exi Blue camera (Qimaging)[86].

### Confocal microscopy

Confocal images were acquired using Zeiss Laser Scanning Confocal Microscope (LSM) 800 with Zen blue software under identical setting and by using Plan-Apochromat 63x/1.40 oil DIC M27 objective, where fixed cells were exposed under laser wavelength 561 nm for Texas Red® ($\lambda_{ex}$ 592 nm; $\lambda_{em}$ 614 nm) and laser wavelength 405 nm for DAPI ($\lambda_{ex}$ 353 nm; $\lambda_{em}$ 465 nm). The images were acquired by randomly selecting at least ten microscopic fields from each slide, and fifteen Z-stack optical sections (1 μm each) were obtained from each image for 3D analysis and image quantification. The Maximum Intensity projection (MIP; compressed Z-Stack) images were created from Z-stack and ImageJ was used to quantify the intracellular and intranuclear puncta (Texas Red®). In the ImageJ program, identical settings were used for each image and in parallel with all the treatments and controls. The "Triangle" threshold and noise reduction were used for all images, and the puncta were obtained and counted by point selection. The binary images of both channels (Texas Red® and DAPI) were obtained and outlined. The puncta count images and outline images were overlaid to count the intracellular and intranuclear puncta in each image. The final numbers of puncta counts were used to plot the graph.

### De novo lipogenesis assays

We used published methods for isotope labeling and purification of cellular lipids[12,87]. Cells (0.5 million per well) were plated in a 6-well plate in complete DMEM medium in triplicate. At 24 h after seeding, cells were washed once with PBS and cultured in serum-free DMEM for 16 h; 5 μCi of [1-14C] acetate (NEC084H001MC, Perkin Elmer, Waltham, MA, USA) per ml was added and the cells were cultured for four more hours. Cells were then washed twice with PBS and trypsinized. Cells were pelleted and resuspended in 0.5 ml of 0.5% Triton X-100. The protein concentration of the lysates was determined for normalization. The lysates were extracted with ice-cold chloroform/methanol (2:1 v/v). After centrifugation at 100 × g for 20 min, the organic phase was collected and air-dried. The radioactivity was

### Table 2 | Antibodies used in this study

| Antibody target | Vendor/source | Vendor catalog # | Dilution |
|---|---|---|---|
| DAXX (for IB) | Bethyl Laboratories | A301-352A | 1:20,000 (IB) |
| DAXX (for IB) | Bethyl Laboratories | A301-353A | 1:20,000 (IB) |
| DAXX (for IP, IB, IF PLA, and ChIP) | The Developmental Studies Hybridoma Bank | PCRP-DAXX-5G11 | Hybridoma supernatant. 1:100 for IB, 1:5 for IF, and 1:2 for PLA |
| DAXX (for IP and IB) | GenScript, rabbit polyclonal | This study | 1:10,000 for IB |
| FASN (for IB and IF) | ProteinTech | 10624-2-AP | 1:20,000 for IB, and 1:700 for IF |
| FASN | Santa Cruz | SC-55580 | 1:20,000 (IB) |
| ACC1 | Cell Signaling Technology | 3676, clone C83B10 | 1:20,000 (IB) |
| ACLY | Cell Signaling Technology | 13390, clone D1X6P | 1:10,000 (IB) |
| ACSS2 | Cell Signaling Technology | 3658, clone D19C6 | 1:20,000 (IB) |
| SREBP2 | Cayman Chemical | 10007663 | 1:10,000 for IB, 1:300 for IF and 1:100 for PLA |
| SREBP2 | BD Biosciences | 557037, clone IgG-1C6 | 1:1000 for IB |
| SREBP1 | Santa Cruz | SC-13551, clone 2A4 | 1:500 (IB) |
| SREBP1 | ProteinTech | 14088-1-AP | 1:3000 for IB, 1:300 for IF and 1:100 for PLA |
| FLAG | Cell Signaling Technology | 14793 | 1:10,000 (IB) |
| FLAG (IB, IP and ChIP) | Millipore-Sigma | F1804, clone M2 | 1:1000 (IB), 2 μg per IP or ChIP experiment |
| GFP | Cell Signaling Technology | 2956 | 1:3000 (IB) |
| PCNA | Epitomics | 2714-1, clone EPR3821 | 1:20,000 (IB) |
| Alpha-tubulin | Millipore-Sigma | T5168, clone B-5-1–2 | 1:50,000 (IB) |
| HSP60 | BD Transduction Laboratories | H99020 | 1:50,000 (IB) |
| Rabbit IgG HRP-linked antibody | Cell Signaling Technology | 7074 | 1:10,000 (IB) |
| Mouse IgG HRP-linked antibody | Cell Signaling Technology | 7076 | 1:10,000 (IB) |
| Normal mouse IgG (for IP/ChIP control) | Santa Cruz | SC-2025 | 2 μg per IP or ChIP experiment |

determined with a liquid scintillation counter (Beckman LS 5000TD). The radioactivity was normalized against protein concentration.

## Liquid chromatography (LC)-mass spectrometry (MS) experiments

For lipid analysis, we used these internal lipid standards: triglyceride (TG 15:0/15:0/15:0 and TG 17:0/17:0/17:0, Sigma-Aldrich), lysophosphatidylcholines (LPC, 17:0 and 19:0), phosphatidylcholines (PC, 17:0/17:0 and 19:0/19:0), phosphatidylethanolamines (PE, 15:0/15:0 and 17:0/17:0), phosphatidylserines (PS, 14:0/14:0 and 17:0/17:0), and phosphatidylglycerols (PG, 14:0/14:0 and 17:0/17:0) (Avanti Polar Lipids, Alabaster, AL). The lipid standards were dissolved in 2:1 (v/v) chloroform/methanol to make a 1000 ppm stock solution and a working 100 ppm standard mix was then prepared by diluting the stock solution with the same solvent mixture. For sample normalization, total protein concentration in each sample was determined using a Qubit 3.0 Fluorometer.

Cell lines with a control vector, an shRNA against an indicated gene, WT DAXX, the DSM mutant, or other specified mutations were cultured with the complete DMEM. When cells grew to ~80% confluency, they were washed twice with PBS and cells were detached using a cell lifter.

Cell pellets were washed twice with 40 mM ammonium formate (AF). The cell pellets were resuspended in 50 μL of AF with a vortex in a glass vial and subjected to high efficient bead beater cell disruption to release intracellular lipids. A small amount of the homogenized cell pellet was taken for Qubit protein concentration determination. Lipids were extracted by adding ice-cold chloroform (2 mL) and methanol (1 mL) along with 20 μL of internal standard mixtures. The extraction mixture was incubated on ice for 1 h with occasional vortex mixing. Finally, 1 mL $H_2O$ was added to the mixture, which was incubated for 10 min with occasional vortex mixing. Samples were then centrifuged at $500 \times g$ for 5 min. The lower phase (organic layer) was collected in a separate glass vial and subjected to dry under nitrogen gas at 30 °C using a dryer (MultiVap, Organomation Associates). Dried samples were reconstituted by adding 50 μL isopropyl alcohol and transferred to a glass LC vial with insert. Samples were loaded to an autosampler at 5 °C.

For analyzing lipids, we ran samples for quality control (QC) in each instrument run. A pooled QC sample (a 25 μL aliquot) for each extraction was injected after analyzing every five samples. The pooled QC sample was run to assess system reproducibility, and a blank (solvent mixture only) was used to flush the column. We did not observe any changes regarding the number of background ions, which always corresponded to the specific solvent used for lipid extraction. Also, we did not notice any effects on the reproducibility of ion source regardless of solvents used for extraction. The stability and repeatability of the instruments were evaluated using identical neat QC samples (a mixture of all internal standards in deuterated form) throughout the process of sample injection. Principal component analysis (PCA) was performed to evaluate the variation of QC samples. All neat QC samples clustered together, confirming the stability and reproducibility of our experimental lipid analysis system.

For data collection, processing, and analysis, we used a Dionex Ultimate 3000 UHPLC system coupled to a Q Exactive™ hybrid quadrupole-orbitrap mass spectrometer operated in HESI-positive and negative ion mode. A Supelco Analytical Titan reverse-phase column (RPC) C18 (2.1 × 75 mm with 1.9 μm monodisperse silica) equilibrated at 30 °C with solvents A (acetonitrile and water 60:40, v/v) and B (isopropyl alcohol, acetonitrile, and water 90:8:2, v/v/v) as mobile phases was used for data collection. The flow rate was 0.5 ml/min, and the injection volume was 5 μL. The total run time was 22 min, including a 2-min equilibration. The MS conditions for positive and negative ion modes were spray voltage at 3.5 kV, sheath gas at 30 arbitrary units, sweep gas at 1 arbitrary unit, auxiliary nitrogen pressure at 5 arbitrary

units, capillary temperature at 300 °C, HESI auxiliary gas heater temperature at 350 °C, and S-lens RF at 35 arbitrary units. The instrument was set to acquire in the mass range of most expected cellular lipids and therefore $m/z$ 100–1500 was chosen with a mass resolution of 70,000 (defined at $m/z$ 200). Global lipid profiling was performed using full scan and ddMS2 (data-dependent MS-MS).

Data were recorded from 0.0 to 17 min as total ion chromatography (TIC) and then corresponding MS data were extracted using Thermo Xcalibur (version 2.2.44). After data collection, raw data files were converted to mzXML format using the Proteowizard MSConvert software. MZmine 2.15 (freeware) was used for mass detection with mass detector centroid noise set at 1.0E5 using only MS level 1 data; chromatogram building and deconvolution were then applied ($m/z$ tolerance, 0.005 or 10 ppm; retention time tolerance, 0.2 min; minimum time span, 0.1 min; and minimum height, 5.0E5) followed by isotope grouping, alignment ($m/z$ tolerance, 0.005 or 10 ppm; retention time tolerance, 0.2 min), and gap filling (m/z tolerance, 0.005 or 10 ppm; retention time tolerance, 0.2 min, and intensity tolerance 25%). MZmine-based online metabolite search engine KEGG, MMCD database, XCMS online database, Metaboanalyst 3.0, R program, and internal retention time library were used for the identification and analysis of metabolites.

## Peptides

The SIM2 (DPEEIIVLSDSD) and the TAMRA (5-carboxytetramethylrhodamine)-SIM2 peptides were synthesized at >95% purity by GenScript (Piscataway, NJ).

## In vivo tumor growth and treatment experiments

All mice were maintained under pathogen-free conditions. Female NSG (NOD.Cg-Prkdc[scid]Il2rg[tm1Wjl]/SzJ) mice, between the ages of 4–12 weeks, were injected subcutaneously in a mammary fat-pad area with one million cells in 100 μl of complete DMEM (MDA-MB-231-derived cell lines) or in a suspension of 50 μl of Matrigel and 50 μl of cell suspension (MDA-MB-468-derived cell lines). The prostate cancer R1-AD1-derived cells (Matrigel suspension) and the colon cancer HCT116-derived cells (medium suspension) were injected subcutaneously in a flank of male NSG mice (the R1-AD1 model) or that of both female and male NSG mice (the HCT116 model). For syngeneic 4T1 and E0771 breast cancer models, female BALB/c and C57BL/6 mice were used, respectively. For the syngeneic prostate cancer TRAMP-C2 model, male NSG mice were used. Tumor growth was monitored by measuring tumor dimensions using a digital caliper once a week until endpoint. Tumor volume was calculated with the formula ½ × length × width². At the endpoint, mice were euthanized, tumors were excised, weighed, and photographed. During experiments, any tumor-bearing mice with tumor size >1.5 cm, tumor ulceration, and a body condition score ≤2 were euthanized. Euthanasia was done using the carbon dioxide inhalation method.

For the in vivo SIM2 treatment study, MDA-MB-231 xenograft tumors were established as above and the mouse mammary tumor cell line 4T1 syngeneic tumors were established by transplanting one million cells in 100 μl of DMEM into mammary fat pads of female BALB/c mice. When tumors grew to a palpable size, tumor-bearing mice were randomized into vehicle and SIM2 treatment arms, so that each group has similar distributions of tumor volumes. The vehicle consisted of 33% (2-hydroxypropyl)-β-cyclodextrin (HPBCD, RND Center Inc., La Jolla) in PBS and 45% polyethylene glycol 400 (Alfa Aesar, Tewksbury, MA), which was filtered through a 0.22 μm filter. The SIM2 peptide was formulated at 5 mg ml⁻¹ in the vehicle. Tumor-bearing mice were injected intraperitoneally with 100 μl of vehicle or the formulated SIM2 peptide once daily every weekday until a predefined endpoint. During the treatment, tumor dimensions and mouse body weights were recorded once weekly. At the endpoint, tumors were excised, weights and photographed. Tumors were flash-frozen in liquid nitrogen, and then stored at −80 °C. Tumor lysates were prepared for

immunoblotting analysis. Animal use has been approved for this project by the University of Florida IACUC.

## ChIP-seq analysis

The panel of MDA-MB-231-derived cell lines (control, WT DAXX and DSM mutant OE) were cultured in complete DMEM. ChIP experiments were performed essentially as described[89]. Briefly, at about 90% confluency, the cells were crosslinked by adding 37% formaldehyde to the final concentration of 1% for 10 min at room temperature. Crosslinking was stopped by adding glycine to the final concentration of 125 mM. Cells were lifted, washed with cold PBS, and pelleted by centrifugation. The cells were resuspended in a swelling buffer in the presence of the protease inhibitor cocktail (Millipore-Sigma P8340) and then pelleted and resuspended in the SDS lysis buffer. The lysates were transferred to a Covaris microTUBE and sonicated with an E220 Covaris Ultrasonicator. Chromatin fragmentation (~500 bps) was verified through agarose gel electrophoresis. The fragmented chromatins were diluted and incubated with a control IgG and the DAXX mAb (5G11) along with protein A/G magnetic beads. The beads were washed sequentially with a low salt buffer, high salt buffer, LiCl buffer, and TE buffer (twice). The immunoprecipitated chromatins were eluted at 65 °C for 15 min, and the eluted chromatins were subjected to proteinase K digestion at 65 °C for 3 h. The DNAs were recovered through a Qiagen mini-prep column. The immunoprecipitated DNAs were used for qPCR and library construction and high throughput sequencing using an Illumina Hi-Seq 2500 sequencer.

## Bioinformatics analysis

The TCGA gene expression and CPTAC protein expression data were analyzed using ULCAN, a cancer OMICS portal (http://ualcan.path.uab.edu/). The TCGA gene expression correlation analysis was conducted using GEPIA (gene expression profiling interactive analysis) portal (http://gepia.cancer-pku.cn/). The Kaplan-Meier survival analysis of TNBC patients based on the TCGA and METABRIC datasets were analyzed as previously described[90]. Briefly, TNBC patients were grouped into high (red) and low (blue) mRNA expression of *DAXX* along with a panel of selected lipogenic genes (TCGA) or *DAXX* mRNA levels alone (METABRIC). Kaplan-Meier analysis based on microarray data of the *DAXX* mRNA expression and recurrence-free survival (RFS) with the "best cutoff" option was conducted at the kmplot.com portal. For Ingenuity Pathway Analysis (IPA), differentially expressed genes (fold-change over ±1.3 and *P* value < 0.05) were used (Ingenuity Systems, Qiagen Bioinformatics, http://www.ingenuity.com). Gene Set Enrichment Analysis (GSEA) was performed using the Java desktop software (http://software.broadinstitute.org/gsea/index.jsp)[88]. The GSEA tool was used in pre-ranked mode with all default parameters. For RNA-seq data analysis, we used an established RNA-seq data analysis pipeline[91]. Briefly, fastq files were aligned to Genome Reference Consortium Human Build 38 (GRCh38) using HISAT2 (version 2.2.1-3n)[92]; the transcripts assembling was performed using StringTie (version v1.3.4)[93] with RefSeq as transcripts ID; and the normalized counts (by FPKM) was called using Ballgown (version v2.12.0)[94]. The differential expression analysis was performed using R package limma (edgeR version 3.38.4)[95]; and the pathway enrichment analysis was performed using IPA. ChIP-seq sequencing reads (Fastq files) were mapped to the human genome (GRCh37/hg19) using Bowtie2[96], where option –local was specified to trim or clip unaligned reads from one or both ends of the alignment. Genome browser BedGraph tracks and read density histograms were generated using SeqMINER. Peak finding and annotation to the nearest Refseq gene promoter were performed and de novo motif discovery was carried out using HOMER[97].

## Statistical analysis and reproducibility

Gene expression assays were conducted in two to three biological replicates. Metabolic profiling assays were performed in four to six replicates. Data are presented as the mean along with standard error of the means (SEM). Unless indicated otherwise, unpaired two-tailed Student's *t* test was used to compare two groups of independent samples using Prism 9. For *P* values calculated using Prism 9, the program setting does not generate the exact values if *P* < 0.0001. Western blot images and fluorescence photomicrographs are shown as representatives of three or more independent experiments with similar results. Uncropped and unprocessed scans of Western blots are provided in the Source Data file.

## Reporting summary

Further information on research design is available in the Nature Portfolio Reporting Summary linked to this article.

## Data availability

The microarray and RNA-seq data generated in this study have been deposited at NCBI under accession codes GSE190596, GSE223583, and GSE192420. The ChIP-seq data generated in this study have been deposited at NCBI under accession code GSE190783. All data are available in the article, Supplementary Information, and source data. Source data are provided with this paper.

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

## Acknowledgements

We thank Shuang Huang and Maria Zajac-Kaye for providing reagents, Subramaniam Shyamalagovindarajan and Ranjan Perera for high throughput sequencing for the ChIP-seq experiments (supported the grant 5BC08 from Bankhead-Coley Cancer Research Program, Florida Department of Health to Perara). This work was supported by grants from Bankhead-Coley Cancer Research Program (4BF02 and 6BC03), and James and Esther King Biomedical Research Program (6JK03, 20K07, 21K03, and 22K05), Florida Department of Health, Florida Breast

Cancer Foundation, and UF Health Cancer Center (to D. Liao). Mass spectrometry-based global lipidomics work was supported by grant from National Institutes of Health (U24DK097209 to T.G.). J. Li was supported by the Intramural Research Program of the NIH, National Institute of Environmental Health Sciences.

## Author contributions

I.M. designed, and performed experiments, analyzed, and interpreted data, and contributed to writing and revision; G.T., J.W., T.E.H., B.J.K., N.A., S.H., C.M., A.M., L.Z., J.E.L., A.W., S.W., J.S., M.L.L., A.C., L.Y.Z., and H.T.P. conducted experiments, acquired and analyzed data; I.M. and J.L.L. performed IPA analysis; I.M. conducted and analyzed the lipidomic data using mass spectrometry under the supervision of T.G.; I.M., H.D., J.L.L., and Z.H. conducted bioinformatics analysis. Y.D. analyzed and interpreted data and contributed to writing; D.L. designed and conducted experiments, analyzed, and interpreted data, acquired funding, supervised the entire study, and wrote the paper.

## Competing interests

I.M., G.T., and D.L. are co-inventor of a US patent application related to this study, which has been filed on behalf of the University of Florida Research Foundation. The remaining authors declare no competing interests.
