## [Peer Review File · Nature Communications]

DAXX drives de novo lipogenesis and contributes to tumorigenesisReviewers' comments:

Reviewer #2 (Remarks to the Author):

This manuscript report the involvement of DAXX in the regulation of transcription of genes involved in lipogenesis. The authors report that DAXX binds to the transcription factors SREBP1/2 via the DHB and HBD DAXX domains. They further showed that DAXX is enriched at promoters of lipogenic genes and DAXX overexpression increased transcription of genes regulated by SREBP1/2. They then involved DAXX expression in tumor growth and showed that DAXX SUMO defective binding mutant failed to promote tumor growth. They then provide evidence that DAXX enhances in vivo tumor growth via SREBP2. Next the authors involved DAXX phosphorylation in the the regulation of its binding with SREBP proteins and further showed the involvement of DAXX phosphorylation in tumor growth in vivo. In the final figure the authors presents evidence that DAXX sim2 peptide blocks DAXX-SREBP interaction and suppress tumor growth in vivo. Overall the paper contains many interesting new discoveries however this study is too preliminary to be considered for publication in Nature Communication. It addresses too many questions and delivers few truly convincing answers. The story about DAXX DSM and DAXX phosphorylation should be the subject of two independent story with more in depth mechanical studies.

Major concerns.

1. The results part of the manuscript is too sketchy in describing the experiments and the rationale behind them due the extent of data presented. Splitting the manuscript in two independent part would allow the authors more space to their experiments.
2. Figure 2 showed that DAXX binds SREBP proteins via 2 domains, ie DHB and HBD. There is no follow up in the paper regarding the implication of this two regions in DAXX function.
3. Following the ChIP seq data, the authors implied that DAXX sumo binding motif are involved in DAXX binding to chromatin while a DAXX sim peptide blocks the interaction with SREBP proteins. Therefore it seems that two different mechanisms are regulated by these motifs, which are not discussed in this manuscript neither further analysed.
4. The authors conclude that DAXX is recruited to the promoters of lipogenic genes by interacting with SREBP2 however a ChIP seq or ChIP qPCR experiments with dowregulation of SREBP proteins showing a lack of DAXX recruitment is missing.
5. Previous studies have shown that DAXX act as both a transcriptional co-repressor and co-activator, and this role is at least in part mediated by the regulation of DAXX phosphorylation (Ecsedy et al., 2003; Boellmann et al., 2004; Santiago et al., 2009; Chang at al. 2011; Michod et al. 2012). Some of the regulators of DAXX phosphorylation have also been described (Ecsedy et al., Chang at al., Michod et al.). The results presented have been barely discussed in light of these previous studies or some of these studies have been omitted in this manuscript.

Minor concerns

1. Fig 2D too many bands are present to conclude anything. The antibody used might not be specific.
2. Fig 3F The reduction of DAXX DSM is not that evident in the figure for some of the region showed (ie ACACA, FASN).

Point-by-point response to the reviewers' comments

RE: NCOMMS-19-03772B

Reviewer #2 (Remarks to the Author):

This manuscript report the involvement of DAXX in the regulation of transcription of genes involved in lipogenesis. The authors report that DAXX binds to the transcription factors SREBP1/2 via the DHB and HBD DAXX domains. They further showed that DAXX is enriched at promoters of lipogenic genes and DAXX overexpression increased transcription of genes regulated by SREBP1/2. They then involved DAXX expression in tumor growth and showed that DAXX SUMO defective binding mutant failed to promote tumor growth. They then provide evidence that DAXX enhances *in vivo* tumor growth via SREBP2. Next the authors involved DAXX phosphorylation in the the regulation of its binding with SREBP proteins and further showed the involvement of DAXX phosphorylation in tumor growth *in vivo*. In the final figure the authors presents evidence that DAXX sim2 peptide blocks DAXX-SREBP interaction and suppress tumor growth *in vivo*. Overall the paper contains many interesting new discoveries however this study is too preliminary to be considered for publication in Nature Communication. It addresses too many questions and delivers few truly convincing answers. The story about DAXX DSM and DAXX phosphorylation should be the subject of two independent story with more in depth mechanical studies.

Response: We are grateful for the very thoughtful review of our manuscript and encouraged by the overall positive comments from this reviewer. We agree that the previous version of this manuscript covered too many different, although interesting, points. We have now focused on the biochemical characterizations of the DAXX-SREBP interactions, the general importance of the DAXX-SREBP axis on lipogenic gene expression, lipogenesis, and tumor growth, as well as roles of the DAXX's SUMO-binding property in these processes. The regulatory roles of DAXX phosphorylation in DAXX-SREBP interactions and lipid production are not covered in the revised manuscript, which will be addressed in a future manuscript.

Major concerns.

1. The results part of the manuscript is too sketchy in describing the experiments and the rationale behind them due the extent of data presented. Splitting the manuscript in two independent part would allow the authors more space to their experiments.

Response: We agree, as discussed above.

2. Figure 2 showed that DAXX binds SREBP proteins via 2 domains, ie DHB and HBD. There is no follow up in the paper regarding the implication of this two regions in DAXX function.

Response: We have addressed this by testing the functional implication of a DAXX mutant with the depletion of amino acids 327-335 within HBD (del 327-335). This construct does not bind SREBP1/2 (Fig. 2D, and F and Supplementary Fig. S6D). This mutant protein is expressed at a similar level as the WT DAXX. However, this mutant was impaired in promoting lipid production compared to wt DAXX (Fig. 3F).

We have also attempted to assess functional effects of another mutant with a mutation (I127A, del 129-132) within DHB (termed 4HB in the revised manuscript). This DAXX mutant also does not bind SREBPs. However, this mutant was produced at a very low level in breast cancer cell lines such as MDA-MB-231, although it expressed well in transfected 293T cells (Fig. 2 and Supplementary Fig. S5D). Therefore, this mutant cannot be used for functional study in breast cancer cells.

3. Following the ChIP seq data, the authors implied that DAXX sumo binding motifs are involved in DAXX binding to chromatin while a DAXX sim peptide blocks the interaction with SREBP proteins. Therefore it seems that two different mechanisms are regulated by these motifs, which are not discussed in this manuscript neither further analysed.

Response: We have provided new data to show that the two DAXX SIMs are both important to bind mature SREBP2 (Fig. 2F in the revised manuscript). These data along with the ChIP-seq data suggest that SUMO-modifications of SREBP1/2 and possibly also core histones are important for chromatin recruitment of DAXX. We have discussed these points in the revised Discuss section (lines 382 – 391).

4. The authors conclude that DAXX is recruited to the promoters of lipogenic genes by interacting with SREBP2 however a ChIP seq or ChIP qPCR experiments with downregulation of SREBP proteins showing a lack of DAXX recruitment is missing.

Response: We have conducted ChIP-qPCR using cells with SREBP2 knockdown. We found that SREBP2 downregulation impaired DAXX's binding to several lipogenic gene promoters such as FASN, ACACA and SREBF2 compared to the control cells (Fig. 4H in the revised manuscript).

5. Previous studies have shown that DAXX act as both a transcriptional co-repressor and co-activator, and this role is at least in part mediated by the regulation of DAXX phosphorylation (Ecsedy et al., 2003; Boellmann et al., 2004; Santiago et al., 2009; Chang et al. 2011; Michod et al. 2012). Some of the regulators of DAXX phosphorylation have also been described (Ecsedy et al., Chang et al., Michod et al.). The results presented have been barely discussed in light of these previous studies or some of these studies have been omitted in this manuscript.

Response: To address this point, we have added a paragraph in the revised Discussion (lines 413 – 426). Since roles of DAXX phosphorylation are no longer covered in the revised manuscript, we have omitted discussions of literature regarding the regulatory functions of DAXX phosphorylation.

Minor concerns

1. Fig 2D too many bands are present to conclude anything. The antibody used might not be specific.

Response: The antibody was extensively validated in our experiments to be highly specific to SREBP2 (see Fig. 2 and Supplementary Fig. S4). The blots for the previous Fig. 2D was an anomaly and has been removed from the revised manuscript.

2. Fig 3F The reduction of DAXX DSM is not that evident in the figure for some of the region showed (ie ACACA, FASN).

Response: The chromatin-binding difference between WT and DSM is indeed relatively small for some genic regions, although the reduction of DSM binding signals for some lipogenic genes (e.g. HMGCS1, MSMO1 and CYP51A1) as well as globally (revised Fig. 4B and C) was quite clear. We do not fully understand this. We have boxed regions with clearer signal reduction in the revised Fig. 4F.

Reviewers' comments:

Reviewer #1 (Remarks to the Author):

This manuscript aims to explore the underlying mechanism of how DAXX is able to regulate lipogenesis in breast cancer cell line. They initially found that knockdown of DAXX dramatically reduced the expression of genes regulating fatty acid and cholesterol synthesis via using microarray assay. Then authors tested the binding between DAXX and SREBP-1 and SREBP-2 via co-IP and claimed that DAXX via interaction with SREBP-1/2 regulates lipogenesis pathway. They further used ChIP-seq to show the binding of DAXX on the promoters of lipogenesis genes. Finally, they used DAXX SIM2 peptide to interfere DAXX SUMO-binding activity and showed that tumor growth inhibition, lipogenesis suppression. Together, they conclude that DAXX through interacting with SREBP-1/2 promotes lipogenesis and tumor growth. It is an interesting study and observation for DAXX involving in lipid metabolism. However, as the previous reviewer pointed, the study remains very preliminary, the quality of many data is very weak and unreliable, and mechanistic study is not deep, which together do not support their conclusion. The authors particularly suffer the reliability of SREBP-2 antibody and the Co-IP results of SREBP-2 and DAXX is poor, which strongly weaken the conclusion.

Major points:

1. Co-IP results for DAXX and SREBP-1 and -2 are not convincing and not realizable. Particularly, the molecular weight of Western blots for SREBP-2 are not consistent between input vs. immunoprecipitants, and are different in different cell lines and different experiments. Endogenous Co-IP blots for SREBP-2 and SREBP-1 with DAXX are very poor and questionable.

1A. In Fig. 2A, Co-IP assay, in MDA-MB-231 cells, first lane shows SREBP-2 input bands are >170Kd, <130 but >100 Kd and a 55 kd mature form. But DAXX putdown band shows 170 KD band, while which was not seen in input lane. Similar in PC3, there is no band over 100kd for SREBP-2 in input lane, while DAXX putdown band is shown between 170 and 130 kd. Authors claim that DAXX can putdown a SREBP-2 band around 170 kD which is posttranslational, while it can not be detected in input lysates. This claim is not convinced and very questionable. Moreover, in same cell line (MDA-MB-231), SREBP-2 precursor size is different between Fig. 2A, 6C, 7A, 7D, further question the data quality and reliability.

1B. Supplemental Fig. S5A, SREBP-1 blot quality is very poor. Particularly, SREBP-1 precursor was seen in both cytosol and nuclear extracts, but not seen in input (particularly in cytosol). This is uncommon results and significantly questionable.

1C, In Fig. S5C, mature SREBP-2 was overexpressed with DAXX in cell line, while DAXX IP mostly putdown the precursor SREBP-2 bands (around 130 kD), not mature band, challenging the conclusion that DAXX can bind to SREBP-2 mature form. In comparison, in input blots, SREBP-2 shows 100 Kd and around 75 bands, while no 130Kd precursor bands shown in IP assay.

2. The study is also very vague for demonstrating the function of DAXX SUMO-binding with SREBP-1 or -2. The only data the manuscript present are the mutation/deletion of DAXX, or SIM peptide interrupt the interaction of DAXX-SREBPs (poor results). It is hard to follow SUMO-binding function for this interaction with SREBPs. Does DAXX mutation impair DAXX SUMOylation which then can't bind to SREBPs? If yes, the study needs to show the changes of DAXX SUMOylation bands. If DAXX interacts with SREBPs through binding to the SUMO site in SREBPs, the specific amino acids in SREBPs for SUMOylation needs to be mutated to prevent SUMOylation, then check whether DAXX can still bind to SREBPs. Direct data show the SUMOylation on DAXX or SREBPs needs to be included to show the evidence.

3. The author claim that DAXX can bind both full-length precursor and mature form of SREBPs, while what is the function of DAXX interacting with precursor of SREBPs leaves vacancy. SREBP precursors traffic from ER to Golgi for cleave and release of mature form. Does DAXX facilitate SREBP trafficking? The co-immunofluorescence staining for both SREBPs and DAXX need to conduct for the co-localization. Fig. 2B PLA assay for their interaction leaves no explanation of how

this assay is performed. SREBP-2 should locate in the ER membrane in the cytoplasm, not like dot shown in "No serum" condition.

Other points:

1. Why authors use different internal controls in this manuscript including PCNA (FIG. 1A), tubulin (Fig. 3C), HSP60 (Fig.5C, 6C, and 7D),DNMT1 (Fig. S2 D)?
2. Fig. 1C, Insig-1 is a direct SREBPs downstream target gene, it is inconsistent that SREBPs levels are inversely correlated with Insig-1 expression.
3. Fig. 1E, RT-qPCR confirmation of DAXX on SREBP and downstream target gene expression is very minor in comparison with microarray data in Fig. 1B, which questions the direct function of DAXX on this genes. Western blot data to show these protein levels in Fig. 1A and other sessions with DAXX KD/OE/DSM should be tested. If SREBPs and their downstream regulated protein levels are not significant changed upon DAXX manipulation, which will overturn the conclusion.
4. Fig. 6C, why the mature band of srebp2 had no big change after knockdown?
5. Fig.3C, some genes are not placed in correct location, such as ACLY should be at Citrate to Acetyl-CoA position, ACAT2 is at Acetyl-CoA to Acetoacetyl-CoA position.
6. Fig. 6. The data do not support the claim that DAXX enhances in vivo tumor growth via SREBP2. Knockdown of SREBP-2 can affect tumor growth without DAXX overexpression. If want to claim that, the study should knock down DAXX first, if SREBP-2 go down, then re-express SREBP-2 to check whether it can restore tumor growth.
7. Fig. S2B and S2C are inconsistent. FASN levels are clearly shown to change in different conditions, while S2C WB band for FASN change is very minor. In addition, SREBP-2 change is very minor in S2C and S2D upon DAXX manipulation, no mature band shown in MDA cell line.
8. Fig. S4. The bands of SREBP-2 are not convincing. Different cell lines have big different for the bands of SREBP-2 full length, which severely question the antibody's reliability. BD SREBP-2 antibody needs to be used to test SREBP-2 bands in different cancer cell lines to see whether the molecular weight of full-length of SREBP-2 is consistent with Cayman antibody.

Reviewer #2 (Remarks to the Author):

The revised manuscript by Mahmud & al is a major improvement. The authors covered all my concerns from the previous version and focused their manuscript. The readability of the manuscript has also improved. I therefore recommend the revised manuscript for publication in Nature Communication.

Point-by-point response to reviewer's comments

RE: NCOMMS-19-03772B

Reviewer #1 (Remarks to the Author):

Reviewer 1's comments: "This manuscript aims to explore the underlying mechanism of how DAXX is able to regulate lipogenesis in breast cancer cell line. They initially found that knockdown of DAXX dramatically reduced the expression of genes regulating fatty acid and cholesterol synthesis via using microarray assay. Then authors tested the binding between DAXX and SREBP-1 and SREBP-2 via co-IP and claimed that DAXX via interaction with SREBP-1/2 regulates lipogenesis pathway. They further used ChIP-seq to show the binding of DAXX on the promoters of lipogenesis genes. Finally, they used DAXX SIM2 peptide to interfere DAXX SUMO-binding activity and showed that tumor growth inhibition, lipogenesis suppression. Together, they conclude that DAXX through interacting with SREBP-1/2 promotes lipogenesis and tumor growth. It is an interesting study and observation for DAXX involving in lipid metabolism. However, as the previous reviewer pointed, the study remains very preliminary, the quality of many data is very weak and unreliable, and mechanistic study is not deep, which together do not support their conclusion. The authors particularly suffer the reliability of SREBP-2 antibody and the Co-IP results of SREBP-2 and DAXX is poor, which strongly weaken the conclusion."

Response: We are glad that this Reviewer thought this is an interesting study. We appreciate the thoughtful critiques. We have provided additional data to address the reviewer's main concern about the data supporting DAXX-SREBP interactions. Additionally, we have provided clarifications on the comments related to immunoprecipitation and immunoblotting data that support the DAXX-SREBP interaction, as detailed below. There are multiple lines of evidence supporting our conclusion that DAXX promotes lipogenesis through interacting with SREBPs to promoting lipogenic gene expression. In addition to co-immunoprecipitation data, the DAXX-SREBP interaction is supported by proximity ligation assay and immunofluorescence co-localization (Supplementary Fig. S6; new data in the revised manuscript). The enrichment of SRE motifs in DAXX-bound chromatin sites (Fig. 4 and Supplementary Fig. S8) and the importance of SREBP2 for DAXX's chromatin association (Fig. 4H) also support the DAXX-SREBP interaction.

Major points:

Reviewer 1's comments: "1. Co-IP results for DAXX and SREBP-1 and -2 are not convincing and not realizable. Particularly, the molecular weight of Western blots for SREBP-2 are not consistent between input vs. immunoprecipitants, and are different in different cell lines and different experiments. Endogenous Co-IP blots for SREBP-2 and SREBP-1 with DAXX are very poor and questionable."

Response: We respectfully disagree with these assessments, as explained below. Further data of co-IP of DAXX and SREBP-2 is provided in the revised manuscript (Fig. 2D).

Reviewer 1's comments: "1A. In Fig. 2A, Co-IP assay, in MDA-MB-231 cells, first lane shows SREBP-2 input bands are >170Kd, <130 but >100 Kd and a 55 kd mature form. But DAXX putdown band shows 170 KD band, while which was not seen in input lane. Similar in PC3, there is no band over 100kd for SREBP-2 in input lane, while DAXX putdown band is shown between 170 and 130 kd. Authors claim that DAXX can putdown a SREBP-2 band around 170 kD which is posttranslational, while it can not be detected in input lysates. This claim is not convinced and very questionable. Moreover, in same cell line (MDA-MB-231), SREBP-2 precursor size is different between Fig. 2A, 6C, 7A, 7D, further question the data quality and reliability."

Response: In Fig. 2A, the corresponding band of the full-length (precursor) SREBP-2 with MW between 170 kDa and 130 kDa, although weak, is clearly detectable in the input lane for both MDA-MB-231 and PC3 cell lines. The isoform of SREBP-2 precursor with MW between 170 and 130 kDa is seen in Fig. 2A, Fig. 6C, Fig. 7A, and 7D.

Reviewer 1's comments: "1B. Supplemental Fig. S5A, SREBP-1 blot quality is very poor. Particularly, SREBP-1 precursor was seen in both cytosol and nuclear extracts, but not seen in input (particularly in cytosol). This is uncommon results and significantly questionable."

Response: We observed that the SREBP-1 precursor form that more strongly binds to DAXX has a MW between 170 kDa and 130 kDa. In Fig. S5A, the Western blot band corresponding to the canonical SREBP-1 precursor with MW between 130 and 100 kDa is more visible, but the larger precursor band that is enriched in the DAXX immunoprecipitates is visible in the Input lane in both cytosolic and nuclear fraction. The legend to Supplementary Fig. S5 clearly describes these patterns. These observations are consistent with data shown in Fig. 7A.

Reviewer 1's comments: "1C, In Fig. S5C, mature SREBP-2 was overexpressed with DAXX in cell line, while DAXX IP mostly putdown the precursor SREBP-2 bands (around 130 kD), not mature band, challenging the conclusion that DAXX can bind to SREBP-2 mature form. In comparison, in input blots, SREBP-2 shows 100 Kd and around 75 bands, while no 130Kd precursor bands shown in IP assay."

Response: We have provided new data that further support the interaction of mature SREBP-2 and DAXX (Fig. 2D). With respect to the data shown previously in Fig. S5C, mature SREBP2 with two FLAG epitope tags in tandem in the N-terminus (F-SREBP2m) and the mouse full-length SREBP-2 were expressed in transfected 293T cells. However, DAXX was not overexpressed. Our data show that the precursor SREBP2 that binds preferentially to DAXX has a MW between 170 kDa and 130 kDa, which was not detectable by Western blot in the input samples of 293T cell extracts except when the full-length SREBP-2 was overexpressed by transfection (Fig. S5C lane 5). Consistent with other data in our manuscript and in the literature, the canonical SREBP-2 precursor has a MW between 130 and 100 kDa in 293T cells (Figs. S4A and S5C). However, the co-IP of the canonical SREBP2 precursor with DAXX was only detectable when the full-length SREBP-2 was overexpressed (Fig. S5C lane 5, IP panel).

Our data show that the endogenous mature form of SREBP-2 in 293T cells has a MW between 55 and 70 kDa, while the transfected mature SREBP-2 has a MW between 70 and 100 kDa (lane 3 in Fig. S5C; also see Fig. 2D and 2G). The endogenous mature SREBP-2 was co-precipitated with DAXX in all samples except when the full-length SREBP-1 was overexpressed (Fig. S5C lane 4). The transfected mature SREBP-2 was clearly co-immunoprecipitated with DAXX (Fig. S5C lane 3, IP panel; also see Fig. 2D and 2G).

Reviewer 1's comments: "2. The study is also very vague for demonstrating the function of DAXX SUMO-binding with SREBP-1 or -2. The only data the manuscript present are the mutation/deletion of DAXX, or SIM peptide interrupt the interaction of DAXX-SREBPs (poor results). It is hard to follow SUMO-binding function for this interaction with SREBPs. Does DAXX mutation impair DAXX SUMOylation which then can't bind to SREBPs? If yes, the study needs to show the changes of DAXX SUMOylation bands. If DAXX interacts with SREBPs through binding to the SUMO site in SREBPs, the specific amino acids in SREBPs for SUMOylation needs to be mutated to prevent SUMOylation, then check whether DAXX can still bind to SREBPs. Direct data show the SUMOylation on DAXX or SREBPs needs to be included to show the evidence."

Response: We have clarified this issue in the revised Discussion (page 17, lines 400-413). Although DAXX and SREBPs undergo SUMOylation at multiple sites, only DAXX contains defined SUMO-binding domains. Therefore, SUMOylation of SREBPs are probably important for DAXX to bind to SREBPs. However, whereas the SIM-SUMO interaction makes a significant contribution to the DAXX-SREBP interaction, other contacts between DAXX and SREBP are also likely critical. Considerable efforts including mutagenesis of SUMOylation sites in SREBPs and perhaps structural studies will be required to fully define the DAXX-SREBP interaction.

Reviewer 1's comments: "3. The author claim that DAXX can bind both full-length precursor and mature form of SREBPs, while what is the function of DAXX interacting with precursor of SREBPs leaves vacancy. SREBP precursors traffic from ER to Golgi for cleave and release of mature form. Does DAXX facilitate SREBP trafficking? The co-immunofluorescence staining for both SREBPs and DAXX need to conduct for the co-localization. Fig. 2B PLA assay for their interaction leaves no explanation of how this assay is performed. SREBP-2 should locate in the ER membrane in the cytoplasm, not like dot shown in "No serum" condition."

Response: We have co-stained DAXX and SREBPs using immunofluorescence microscopy. Co-localization of DAXX and SREBP1 was observed in the nucleus as well as in the cytoplasm in the culture medium with or without serum in MDA-MB-231 cells (Supplementary Fig. S6). We have expressed GFP-DAXX by transient transfection in MDA-MB-231 cells. Notably, the SREBP1 fluorescence signal intensity was markedly increased

in cells expressing GFP-DAXX compared to cells without GFP-DAXX expression in medium with full serum. Cells without GFP-DAXX expression displayed equal distribution of SREBP1 in the nucleus and cytoplasm, while cells with GFP-DAXX expression exhibited 2-fold higher level of nuclear SREBP1 than that of cytoplasmic SREBP1 (Supplementary Fig. S6A), suggesting that DAXX may promote SREBP1 nuclear translocation. These observations are described in lines 181-194 (pages 8 and 9) in the revised text.

We have described how PLA assay was performed (page 7, lines 158-159, page 27, lines 601-607). Our data suggest that the DAXX-SREBP interaction as detected by PLA may not just occur on ER. We have also discussed possible functional ramification of the interaction between DAXX and precursor SREBPs in the revised Discussion (lines 460-493, pages 19 and 20).

Other points:

Reviewer 1's comments: "1. Why authors use different internal controls in this manuscript including PCNA (FIG. 1A), tubulin (Fig. 3C), HSP60 (Fig.5C, 6C, and 7D),DNMT1 (Fig. S2 D)?"

Response: Although different loading controls were shown, we made sure that equal amounts of total cellular proteins were loaded in each lane. For Western blotting experiments to compare protein abundance in cell cultures (Figs. 1A, 3C, 6C, S2C and S2D), we seeded equal number of cells for each sample. For tumor extracts (Figs. 5C and 7D), protein concentrations were determined by the Bradford method and equal amounts of total proteins per lane were loaded.

Reviewer 1's comments: "2. Fig. 1C, Insig-1 is a direct SREBPs downstream target gene, it is inconsistent that SREBPs levels are inversely correlated with Insig-1 expression."

Response: Fig. 1C shows activated or inhibited "upstream regulator" in IPA. Fig. 1C depicts that the INSIG1 "pathway" is activated in cells with DAXX knockdown, not an indication of increased INSIG1 expression. The depiction reflects the fact that the SREBP lipogenic pathways are repressed when DAXX is depleted, as INSIG1 is a negative regulator of the SREBP pathway. Although not shown, DAXX knockdown resulted in a marked downregulation of INSIG1, again consistent with our conclusion that DAXX regulates lipogenesis through SREBPs.

Reviewer 1's comments: "3. Fig. 1E, RT-qPCR confirmation of DAXX on SREBP and downstream target gene expression is very minor in comparison with microarray data in Fig. 1B, which questions the direct function of DAXX on this genes. Western blot data to show these protein levels in Fig. 1A and other sessions with DAXX KD/OE/DSM should be tested. If SREBPs and their downstream regulated protein levels are not significant changed upon DAXX manipulation, which will overturn the conclusion."

Response: The gene expression pattern shown in Fig. 1B is based on Z-scores, not fold-changes, while Fig. 1E shows relative fold-changes. For most lipogenic genes, the relative fold-changes range from ± 1.5 to ± 2.5 in comparison to control in our microarray data, which is similar to RT-qPCR data. Immunoblotting validation of the protein levels for representative lipogenic genes is provided in Fig. S2C and S2D. We have added the immunofluorescent microscopic image panels of the FASN protein in MDA-MB-231 with DAXX DSM mutant overexpression in the revised Fig. S2B.

Reviewer 1's comments: "4. Fig. 6C, why the mature band of sreb2 had no big change after knockdown?"

Response: One possible explanation is that the precursor and the mature SREBP2 have different metabolic stability. It was shown that mature SREBP1 and SREBP2 are acetylated by p300/CBP, which stabilizes SREBPs (PMID: 12640139). It is possible that mature SREBPs might be more stable.

Reviewer 1's comments: "5. Fig.3C, some genes are not placed in correct location, such as ACLY should be at Citrate to Acetyl-CoA position, ACAT2 is at Acetyl-CoA to Acetoacetyl-CoA position."

Response: This has been corrected.

Reviewer 1's comments: "6. Fig. 6. The data do not support the claim that DAXX enhances in vivo tumor growth via SREBP2. Knockdown of SREBP-2 can affect tumor growth without DAXX overexpression. If want to claim that, the study should knock down DAXX first, if SREBP-2 go down, then re-express SREBP-2 to check whether it can restore tumor growth."

Response: Our observations that SREBP2 knockdown reduces DAXX's chromatin-binding to lipogenic gene promoters (Fig. 4H) as well as WT DAXX OE-mediated tumor growth provide a strong support for an important role for SREBP2 in DAXX's oncogenic function. Nonetheless, we agree that DAXX could promote oncogenesis through other mechanism(s) in addition to the DAXX-SREBP2 axis. We have modified our description accordingly (lines 317, 326 and 1105).

Reviewer 1's comments: "7. Fig. S2B and S2C are inconsistent. FASN levels are clearly shown to change in different conditions, while S2C WB band for FASN change is very minor. In addition, SREBP-2 change is very minor in S2C and S2D upon DAXX manipulation, no mature band shown in MDA cell line."

Response: We have added new immunofluorescence microscopic images of FASN in MDA-MB-231 cells overexpressing the DSM mutant (revised Fig. S2B). The new data show that the DSM mutant was unable to promote FASN expression compared to WT DAXX, consistent with other data demonstrating the inability of the DSM mutant to promote lipogenesis and tumorigenesis (Figs 1, 3, 5 and S7). We have also added new immunoblotting data for SREBP2 (both mature and FL), FASN and ACC1 (revised Fig. S2C). While the increased protein level for FASN appears moderate compared to immunofluorescence data in cells with WT DAXX OE, this has been consistently observed in multiple immunoblotting experiments. It should be noted that the methods of protein detection are fundamentally different between immunoblotting and immunofluorescence microscopy. For example, the epitopes that can be bound by an antibody are likely different in the two methods.

Reviewer 1's comments: "8. Fig. S4. The bands of SREBP-2 are not convincing. Different cell lines have big different for the bands of SREBP-2 full length, which severely question the antibody's reliability. BD SREBP-2 antibody needs to be used to test SREBP-2 bands in different cancer cell lines to see whether the molecular weight of full-length of SREBP-2 is consistent with Cayman antibody."

Response: Multiple lines of evidence show that the Cayman SREBP-2 is specific to SREBP2. This antibody was validated in cells transfected with mature SREBP2 (Figs. 2D, S4A lane 2, and S5C lane 3) and full-length SREBP2 (Fig. S5C lane 5). Additionally, this antibody was validated by SREBP2 knockdown (Fig. 6C). This antibody also consistently detected the endogenous precursor and mature SREBP2. As stated above and in the manuscript, in the cancer cell lines we have tested, the major band of the precursor SREBP-2 has a MW between 130 and 170 kDa in immunoblotting using the Cayman antibody, while a band with a MW between 100 and 130 kDa, presumably corresponding to the unmodified SREBP2 precursor, was detectable in most immunoblots (Figs. 2A, 6C, 7D, S2C, S2D, S4B, S4C). In contrast, in 293T cells, the major visible SREBP2 precursor band has a MW between 100 and 130 kDa (Figs. S4A and S5C). This antibody also consistently detects the mature SREBP2 at 55 kDa in cancer cell lines in immunoblotting (Figs. 2A, 6C, 7D, S2C, S4B, and S4C). The BD SREBP-2 antibody recognizes an epitope outside of the mature form of SREBP2, as documented in the manufacturer's product description. Indeed, it did not bind to the transfected mature SREBP2 (Fig. S4A, lane 2).

Reviewer #2 (Remarks to the Author):

Reviewer #2's comments: "The revised manuscript by Mahmud & al is a major improvement. The authors covered all my concerns from the previous version and focused their manuscript. The readability of the manuscript has also improved. I therefore recommend the revised manuscript for publication in Nature Communication."

Response: We would like to express our appreciation for the very thoughtful and constructive critiques of this reviewer that have been important for us to improve the manuscript.

Reviewers' comments:

Reviewer #3 (Remarks to the Author):

First of all, the authors do not specify which antibody was used for which experiment. Both SREBP2 antibodies are listed as used for immunoblotting. This information should be provided to assess specificity. I have not used any of the antibodies listed there so I cannot comment on their specificity. As a matter of fact, neither of the SREBP2 antibodies used in this study have a literature reference listed on their website.

Regarding the molecular weight of full length SREBP2, it is very difficult to understand why such different sizes are observed. The authors could address post-translational modification (in particular ubiquitylation), which could explain this shift in apparent molecular weight. However, given the difficulty with antibody specificity, the authors should provide a single experiment confirming the specificity of their antibodies using a knock-down or knockout strategy.

Specific comments:

Figure 2A: The molecular weight of full length SREBP2 as indicated is far too large. Full length SREBP2 has a predicted molecular weight of 124 kDa. This is very different from the indicated >170 kDa in this figure. Here, specificity should be demonstrated by silencing of SREBP2.

Figure 2B: The PLA only shows interaction between SREBP2 and DAXX based on signal differences between low or high serum. It is possible that serum alters the background of the PLA detection. The specificity of Co-IPs or PLAs should be confirmed by a silencing strategy targeting either DAXX or SREBP2.

Figure 2D: The expression levels of flag-tagged mature SREBP2 strongly differ between DAXX wt and W622F overexpressing cells. Based on this difference, the reduced binding observed in the DAXX mutant could just reflect this difference. In addition, this difference could indicate that DAXX affects the stability of mature SREBP2, which would also explain the regulation of lipid metabolism genes.

Figure S5A: Here, the authors immunoprecipitate endogenous SREBP2 with DAXX from cytoplasmic and nuclear lysates. While there is some binding to the mature for detect (indicated by the fussy band at 50 kDa), this cannot be interpreted as "strong affinity" as stated in the figure legend.

Page 20 and Figure S6: I have an issue with the statements regarding the cytoplasmic to nuclear translocation of SREBP2 by DAXX. I cannot detect any change in the localisation of SREBP2 in DAXX positive cells in Figure S6. Also, the statements regarding localisation of full length and mature SREBP on page 20 are not consistent with the literature. SREBP2 is processed in response to low sterol concentrations (i.e. low serum) due to the induction of ER to Golgi translocation and the subsequent cleavage by site 1 and site 2 proteases. This process is absent when mature SREBP is expressed, so this is expected to be nuclear. I don't think that the authors can conclude that the localisation of SREBP2 is cytoplasmic, in the absence of ultra-centrifugation-based fractionation or EM imaging.

Page 20: the interpretation of the data of Peterson et al (Reference 18) is slightly wrong. This study has shown that mTORC1 activation prevents the accumulation of mature SREBP in the nuclear periphery by lipin 1. This does not affect nuclear entry, rather sequesters SREBP in a subnuclear compartment that is not accessible to DNA.

Conclusion:

I only looked at some aspects of this manuscript. However, I have the feeling that there is quite a lot of overinterpretation of the data and some technical flaws that need to be clarified before this manuscript can be considered for publication. In my opinion, the authors have not proven that there is a direct link between SREBP and DAXX in driving cancer growth. I would generally side with reviewer 1 on the criticism of this manuscript.

Reviewer #3 (Remarks to the Author):

First of all, the authors do not specify which antibody was used for which experiment. Both SREBP2 antibodies are listed as used for immunoblotting. This information should be provided to assess specificity. I have not used any of the antibodies listed there so I cannot comment on their specificity. As a matter of fact, neither of the SREBP2 antibodies used in this study have a literature reference listed on their website.

Regarding the molecular weight of full length SREBP2, it is very difficult to understand why such different sizes are observed. The authors could address post-translational modification (in particular ubiquitylation), which could explain this shift in apparent molecular weight. However, given the difficulty with antibody specificity, the authors should provide a single experiment confirming the specificity of their antibodies using a knock-down or knockout strategy.

Response: We truly appreciate these insightful comments.

The specificity of the rabbit SREBP2 polyclonal antibody (Cayman Chemical catalog # 1007663) that we have used in this study has been confirmed by shRNA-mediated knockdown of SREBP2 using the TRCN0000020667 lentiviral clone (Fig. 6g; this SREBP2 shRNA has also been validated in a peer-reviewed publication (PMID: 26028026). Additionally, this antibody was also validated in transfection overexpression experiments of the mature SREBP2 (Fig. S6a, lane 3, denoted with a cyan arrow) and full-length SREBP2 (Fig. S6a, lane 5). Notably, this SREBP2 antibody is from the same source as the Abcam anti-SREBP2 antibody (catalog # ab30682). The product information for this antibody from both vendors is appended in this document. As described in the product information, the immunogen for producing this antibody corresponds to the amino acid residues 455-469 of SREBP2, which is within the mature form of SREBP2. Like what we have observed in several cancer cell lines including MDA-MB-231, PC3, Hs587t and HCT116, the product information indicates that the MW of the major band that presumably corresponds to the cleaved N-terminal fragment is 55 kDa. This 55-kDa protein was co-immunoprecipitated with endogenous DAXX (Fig. 2b). Cell fractionation data show that this 55-kDa protein is exclusively nuclear (Fig. S5a and b). Similarly, a transfected N-terminal SREBP2 construct corresponding to the mature SREBP2 also interacts with DAXX in co-IP experiments (Fig. 2e and Fig. S6b).

With respect to the size of the full-length SREBP2 that was co-immunoprecipitated with DAXX, a band of ~170 kDa was also detected with a mouse monoclonal antibody against SREBP2 (BD Biosciences, catalog # 557037) in our new IP experiments (revised Fig. 2a), which confirms our previous data. Co-immunoprecipitation of the ~170 kDa SREBP2 band with DAXX was also observed using 293T cell lysates (Fig. S6a). Notably, the apparent MW of the SREBP2 precursor (FL) in breast cancer cell lines (MDA-MB-231, MDA-MB-468, and Hs578t), colon cancer cell line HCT116, and prostate cancer cell line R1-AD1 (a subline of CWR-R1) is similar (~170 kDa, Fig. S5). Importantly, this form is the major band in the cytoplasmic fraction (Fig. S5a and b). These observations together indicate that the 170 kDa band is likely a bona fide form of SREBP2.

It is noted in the Cayman Chemical product information that the apparent MW of SREBP2 on SDS-PAGE may be higher than the calculated MW (~126 kDa) due to glycosylation of the protein (ref.

PMID: 7493979). Published literature also documented that the apparent MW of the full-length SREBP2 can be larger than the predicted MW, depending on cell types. For example, Mai et al. reported that the MW of the full-length SREBP2 is 165 kDa (PMID: 35833708). Pandyra et al. also showed that the apparent MW of the precursor SREBP2 is significantly larger than 130 kDa (PMID: 26353928). Therefore, the precise MW of the full-length and mature forms of SREBP2 varies in different cell types as well as in the same cell line under different conditions.

Regardless, our data show that DAXX interacts preferentially with the 170 kDa species of SREBP2 precursor, which is probably a posttranslationally modified form of the canonical 125 kDa FL SREBP2. The 170-kDa band was not detected by antibodies against SUMO1, SUMO2/3, mono-ubiquitin, K48-, or K63-linked poly ubiquitin chain. Future studies will be required to determine what type(s) of modifications that promote DAXX to interact with SREBP2 precursor.

To further validate the SREBP-1/2 antibodies used in this study, we have conducted additional experiments to confirm the specificity of the anti-SFREBP1 (ProteinTech, catalog # 14088-1-AP) and anti-SREBP2 (BD Biosciences, catalog # 557037) antibodies using shRNA knockdown (Fig. S5d). These shRNAs have been validated in published literature (the SREBP1 shRNA TRCN0000020607 in PMID: 32568455, and the SREBP2 shRNA TRCN0000020667 in PMID: 26028026).

These antibodies were used in IP, PLA and IF experiments and their use is specifically indicated in relevant figure legends. Notably, the immunogen used for producing the anti-SREBP2 mouse monoclonal antibody was based on the sequence of amino acids 833-1141 of the human SREBP2. Presumably, the smaller form of SREBP2 denoted in Fig. 2a and Fig. S5d represents the C-terminal fragment after cleavage of the precursor (full-length) SREBP2 in the Golgi apparatus. The fragment of SREBP2 is labeled as “C” in the revised manuscript. Nonetheless, it is clear that the SREBP2 shRNA depleted both precursor and the cleaved form of the SREBP2 (Fig. S5d).

All the SREBP1 and SREBP2 antibodies used in our study have also been cited in numerous peer-reviewed publications. The citations for the rabbit polyclonal anti-SREBP2 antibody (Cayman Chemical catalog # 1007663 or Abcam ab30682) can be found at <https://www.citeab.com/antibodies/2864481-10007663-srebp-2-polyclonal-antibody?des=fab8abf3a253a277>; <https://www.abcam.com/srebp2-antibody-ab30682.html>.

The citations for the rabbit polyclonal anti-SREBP1 antibody (ProteinTech, catalog # 14088-1-AP) can be found at <https://www.ptglab.com/products/SREBF1-Antibody-14088-1-AP.htm>

Selected citations for the mouse monoclonal anti-SREBP2 antibody (BD Biosciences, catalog # 557037) include PMID: 32322062 et etc. Additional citations can be found at <https://www.citeab.com/antibodies/2410727-557037-bd-pharmingen-purified-mouse-anti-srebp-2>

The citations for the mouse monoclonal anti-SREBP1 antibody (2A4, catalog # SC-13551, Santa Cruz Biotechnology) can be found at <https://www.scbt.com/p/srebp-1-antibody-2a4>

Specific comments:

Figure 2A: The molecular weight of full length SREBP2 as indicated is far too large. Full length SREBP2 has a predicted molecular weight of 124 kDa. This is very different from the indicated >170 kDa in this figure. Here, specificity should be demonstrated by silencing of SREBP2.

Response: as stated above, we have repeated the IP experiments and used the validated SREBP2 antibody (BD Biosciences, catalog # 557037) to detect co-immunoprecipitated SREBP2. The precursor (full-length) SREBP2 that was co-immunoprecipitated with DAXX has an apparent MW of ~170 kDa, similarly to the results of previous IP experiments (revised Fig. 2b). The specificity of this large SREBP2 band has been validated (Fig. 6g) as stated above.

Figure 2B: The PLA only shows interaction between SREBP2 and DAXX based on signal differences between low or high serum. It is possible that serum alters the background of the PLA detection. The specificity of Co-IPs or PLAs should be confirmed by a silencing strategy targeting either DAXX or SREBP2.

Response: we have repeated the PLA assays using MDA-MB-231 cells with a control vector or a DAXX or SREBP2 shRNA (TRCN0000020667) in serum-free medium using the same antibody pairs as in Fig. 2c. The new data show that no DAXX-SREBP2 PLA puncta were detected in cells depleted of DAXX or SREBP2 (see the figure below):

Figure 2D: The expression levels of flag-tagged mature SREBP2 strongly differ between DAXX wt and W622F overexpressing cells. Based on this difference, the reduced binding observed in the DAXX mutant could just reflect this difference. In addition, this difference could indicate that DAXX affects the stability of mature SREBP2, which would also explain the regulation of lipid metabolism genes.

Response: we appreciate the careful comments. This panel is removed from the revised Fig. 2. Our new (Fig. 2a and Fig. S6b) and previous data (Fig. 2b and Fig. S6a) clearly show that DAXX interacts with SREBP2 (mature and full-length). Whether or not DAXX affects SREBP2 protein stability can be addressed in future study.

Figure S5A: Here, the authors immunoprecipitate endogenous SREBP2 with DAXX from

cytoplasmic and nuclear lysates. While there is some binding to the mature for detect (indicated by the fuzzy band at 50 kDa), this cannot be interpreted as “strong affinity” as stated in the figure legend.

Response: this was an immunoprecipitation experiment to show DAXX-SREBP1 interaction. The DAXX-SREBP1 interaction data are now shown in the revised main Fig. 2. We have conducted new IP experiments. The new data are shown in the revised Fig. 2a. As such, the original panel was removed from the revised Fig. S6. Additional evidence of the interaction of DAXX with the precursor (FL) and mature SREBP1 can be found in Fig. S6a and b in the revised manuscript.

Page 20 and Figure S6: I have an issue with the statements regarding the cytoplasmic to nuclear translocation of SREBP2 by DAXX. I cannot detect any change in the localisation of SREBP2 in DAXX positive cells in Figure S6. Also, the statements regarding localisation of full length and mature SREBP on page 20 are not consistent with the literature. SREBP2 is processed in response to low sterol concentrations (i.e. low serum) due to the induction of ER to Golgi translocation and the subsequent cleavage by site 1 and site 2 proteases. This process is absent when mature SREBP is expressed, so this is expected to be nuclear. I don't think that the authors can conclude that the localisation of SREBP2 is cytoplasmic, in the absence of ultra-centrifugation-based fractionation or EM imaging.

Response: we appreciate the comments. The difference in nuclear SREBP1 abundance between cells expressing transfected GFP-DAXX and not were small and not obvious at first glance. We have quantitatively analyzed the protein abundance of nuclear SREBP1 over numerous transfected cells and untransfected cells and the difference of nuclear SREBP1 levels was statistically significant (revised Fig. S7). Additionally, we obtained new data indicating that DAXX OE increased the level of mature SREBP1 using immunoblotting (revised Fig. S7d).

We have revised the relevant parts of the text to address and reflect these concerns (see the revised texts in pages Page 18 lines 421-423).

Page 20: the interpretation of the data of Peterson et al (Reference 18) is slightly wrong. This study has shown that mTORC1 activation prevents the accumulation of mature SREBP in the nuclear periphery by lipin 1. This does not affect nuclear entry, rather sequesters SREBP in a subnuclear compartment that is not accessible to DNA.

Response: thank you for this insight. We have modified the relevant texts based on this comment (Page 3, lines 65 and 66, Page 18 lines 431-433).

Conclusion:

I only looked at some aspects of this manuscript. However, I have the feeling that there is quite a lot of overinterpretation of the data and some technical flaws that need to be clarified before this manuscript can be considered for publication. In my opinion, the authors have not proven that there is a direct link between SREBP and DAXX in driving cancer growth. I would generally side with reviewer 1 on the criticism of this manuscript.

Response: we are thankful for these thoughtful and constructive critiques. We have provided new validation for SREBP1 and SREBP2 antibodies and conducted additional IP and PLA experiments. Regarding the connection between DAXX and SREBP in driving tumor growth, we have conducted new experiments and obtained a significant amount of new supporting data. Importantly, we have shown that a DAXX mutant with a small deletion that disrupts the DAXX-SREBP1/2 interaction was impaired to promote lipogenesis and tumor growth compared to wt DAXX (revised Fig. 7). Regarding the criticism on overinterpretation, we have made our best efforts to moderate our description to minimize the impression of overinterpretation throughout the text of the revised manuscript.

REVIEWERS' COMMENTS

Reviewer #3 (Remarks to the Author):

The authors have improved the manuscript substantially. They now add high stringency data to confirm the interaction between DAXX and SREBP1/2. They also conducted additional lipidomic analysis to confirm deregulated lipid metabolism in DAXX driven tumours. Most importantly, they provide additional data showing that blocking the interaction between DAXX and SREBP1/2 using a peptide substantially reduces tumour growth. Together, these new data add additional support to their conclusion that DAXX drives lipid synthesis via binding to SREBP1/2 and that this interaction is an important mechanism for tumour growth.